# Digital Alchemy: The Rise of Machine and Deep Learning in Small-Molecule Drug Discovery

**DOI:** 10.3390/ijms26146807

**Published:** 2025-07-16

**Authors:** Abdul Manan, Eunhye Baek, Sidra Ilyas, Donghun Lee

**Affiliations:** 1Department of Molecular Science and Technology, Ajou University, Suwon 16499, Republic of Korea; mananriaz012@gmail.com; 2RexSoft Inc., 1 Gwanak-ro, Gwanak-gu, Seoul 08826, Republic of Korea; 1st-balloon@hanmail.net; 3Department of Herbal Pharmacology, College of Korean Medicine, Gachon University, 1342 Seongnamdae-ro, Sujeong-gu, Seongnam-si 13120, Republic of Korea

**Keywords:** artificial intelligence, drug discovery, big data, virtual screening, machine learning, deep learning

## Abstract

This review provides a comprehensive analysis of the transformative impact of artificial intelligence (AI) and machine learning (ML) on modern drug design, specifically focusing on how these advanced computational techniques address the inherent limitations of traditional small-molecule drug design methodologies. It begins by outlining the historical challenges of the drug discovery pipeline, including protracted timelines, exorbitant costs, and high clinical failure rates. Subsequently, it examines the core principles of structure-based virtual screening (SBVS) and ligand-based virtual screening (LBVS), establishing the critical bottlenecks that have historically impeded efficient drug development. The central sections elucidate how cutting-edge ML and deep learning (DL) paradigms, such as generative models and reinforcement learning, are revolutionizing chemical space exploration, enhancing binding affinity prediction, improving protein flexibility modeling, and automating critical design tasks. Illustrative real-world case studies demonstrating quantifiable accelerations in discovery timelines and improved success probabilities are presented. Finally, the review critically examines prevailing challenges, including data quality, model interpretability, ethical considerations, and evolving regulatory landscapes, while offering forward-looking critical perspectives on the future trajectory of AI-driven pharmaceutical innovation.

## 1. Introduction and Scope

### 1.1. Traditional Drug Discovery Challenges

The journey from initial discovery to market approval for a new drug is notoriously arduous, typically spanning over a decade, often 10–15 years, and incurring costs estimated at approximately USD 2.6 billion [1]. This prolonged timeline and exorbitant investment are compounded by significant bottlenecks in the preclinical phase, which alone can consume 3–6 years and USD 1–6 million. Despite this substantial commitment of resources, only 10% of drug candidates successfully transition from preclinical development to clinical trials [2]. This paradigm, known as Eroom’s law, observes that despite technological advances, the real-world efficiency of pharmaceutical R&D continues to decline, with development costs doubling approximately every nine years. The pharmaceutical industry faces a nearly 90% failure rate for drugs entering clinical trials, primarily due to lack of efficacy, often attributable to the limited predictive power of in vitro models and poor translatability of traditional animal model data to human outcomes. To overcome these bottlenecks, researchers have turned to high-throughput in vitro assays and computer-driven in silico techniques to optimize candidates before costly animal or human studies. Historically, drug discovery relied heavily on empirical trial-and-error methods, a paradigm that began shifting towards more rational drug design (using scientific knowledge to design molecules) with the advent of early computational approaches.

The profound inefficiencies and high attrition rates endemic to traditional drug discovery, manifested as long timelines, escalating costs, and frequent clinical failures, have created a pressing demand for more efficient and predictive early-stage methodologies. This urgent need directly has catalyzed the widespread adoption and investment in AI and ML technologies. The conventional drug discovery process, with its substantial resource drain and low success rates, has inherently generated an imperative for transformative alternatives. AI and ML models promise to reduce timelines and costs by predicting target engagement, toxicity, and pharmacokinetics in silico [3]. The emergence of AI offers a promising avenue for research scientists to process, analyze, and interpret complex data [4]. The extensive application of these technologies represents the future for drug discovery, promising to overcome the limitations inherent in traditional methodologies.

### 1.2. ML and DL: A Paradigm Shift in Drug Discovery

The integration of ML and DL is fundamentally reshaping drug discovery by significantly enhancing data analysis capabilities and predictive accuracy, thereby promising faster and more effective therapeutic development. This transformative momentum is fueled by the convergence of increasingly accessible relevant data, exponential growth in computing power, and profound advancements in deep learning algorithms. ML and DL technologies are poised to dramatically reduce human workload, improve precision, and compress development timelines across the entire pharmaceutical value chain. The historical shift from empirical random screening to rational drug design is being profoundly accelerated and refined by ML and DL, propelling the field toward the de novo engineering of compounds with highly specific and optimized properties. This signifies a fundamental evolution in how rational drug design is conceptualized. Instead of primarily assisting in the identification of promising compounds from pre-existing chemical libraries, ML and DL, particularly through generative models, enable the creation of entirely novel compounds from scratch, designed with specific, pre-defined properties in mind. This represents more than an incremental improvement in efficiency; it signifies a qualitative shift in the very essence of rational drug design, transitioning from an approach focused on identifying the best available option to one centered on synthesizing the optimal desired entity. This fundamentally redefines the starting point and strategic approach to drug discovery.

The emerging lab-in-a-loop concept represents the development of a closed-loop, self-improving drug discovery ecosystem. In this model, AI algorithms are continuously refined using real-world experimental and clinical data. These models generate predictions for drug targets and therapeutic molecules, which are then experimentally validated. The resulting data are fed back to retrain and enhance the models, creating a continuous, self-correcting feedback cycle. This suggests a future in which drug development becomes increasingly autonomous, adaptive, and exponentially accelerating process. This is not merely AI serving as a static tool to assist human researchers; it represents AI’s capacity to learn and improve dynamically based on empirical outcomes. This fundamental shift transforms the drug discovery paradigm from a largely linear, human-driven process (albeit with AI tools) into a cyclical, AI-driven process with human oversight, promising compounding improvements in efficiency, accuracy, and innovation over time.

This review systematically explores the foundational principles and inherent limitations of traditional small-molecule drug discovery methods. It provides a comprehensive analysis of how cutting-edge machine learning (ML) and deep learning (DL) techniques directly address these long-standing challenges, outlining their applications across the drug discovery pipeline, presenting compelling real-world case studies, and discussing prevailing challenges and future directions in this rapidly evolving field (Figure 1).

### 1.3. AI Drug Discovery: From Big Data to Smart Data

The landscape of pharmaceutical research and development has been fundamentally reshaped by the emergence of big data, a concept referring to datasets characterized by huge volume (sheer quantity), velocity (speed of generation), variety (heterogeneity of types), veracity (accuracy and reliability), and value (actionable insights) that traditional tools struggle to manage and analyze (Figure 2). This paradigm shift has been largely propelled by the advent of high-throughput technologies such as microarrays, next-generation sequencing (NGS), mass spectrometry (MS), single-cell omics, and automated screening that have flooded drug discovery with vast, heterogeneous datasets of gene expression, protein abundance, and compound activity characterized by massive volume, rapid velocity, diverse variety, uncertain veracity and potential value at lower costs [5]. The velocity of data inflow now approaches real-time, driven by continuous patient monitoring via wearable sensors and automated high-throughput screening (HTS) pipelines, demanding near-real-time processing and analytics [6]. The inherent variety of data encompasses multiomics (genomics, transcriptomics, epigenomics, proteomics, metabolomics), cheminformatics (chemical structures, bioassays), electronic health records (EHRs), imaging modalities, and text-mined literature associations. Managing this heterogeneity requires specialized preprocessing and integration strategies to harmonize diverse sources, a significant analytical challenge that can also introduce biases if not carefully addressed. Ensuring veracity—the accuracy and reliability of these data—involves correcting for batch effects and annotation errors, particularly in omics experiments, to prevent systematic biases that can compromise model validity. Ultimately, the goal is to transform these raw data into actionable value, such as prioritizing lead compounds, predicting off-target effects, and identifying novel target hypotheses, optimized lead compounds, and patient-stratification biomarkers.

Early AI efforts followed a “more is better” philosophy, accumulating millions of assay results, omics datasets, and combinatorial libraries in an attempt to cover the vast theoretical chemical space of 10^60^–10^80^ compounds. This approach quickly encountered diminishing returns such as noisy labels, batch effects, redundant compounds, and soaring computational costs, yielding models that overfitted and seldom generalized. By refocusing on smart data, researchers now ask not how much data can be collected, but how informative each measurement is. Curated, orthogonal assays and richly annotated metadata allow AI systems to correct for variability, prioritize mechanistically relevant experiments, and learn underlying biochemical principles from far fewer examples. The integration of AI with diverse, high-quality biomedical data enables robust data analysis and informed decision-making, accelerating the drug discovery process (Figure 3). Techniques such as active learning and Bayesian optimization guide iterative experimentation; transfer learning and meta-learning use pretrained models to jump-start new projects; few-shot models extrapolate activity from minimal labels; and generative augmentation simulates novel compounds without costly wet-lab screens [7]. This data-efficient paradigm shifts resources from indiscriminate screening to precision triaging, enabling smaller, multidisciplinary teams to accelerate lead selection, reduce attrition, and democratize AI-driven discovery across both mainstream and niche targets.

### 1.4. Molecular Representations

Choosing effective molecular representations remains a key challenge. Common approaches include SMILES strings (linear notations that serve as a chemical language), which can be processed using sequence-based models such as natural language processing (NLP), recurrent neural networks (RNN), long short-term memory (LSTM), and transformers like ChemBERTa to automatically learn chemical syntax and substructure patterns [8,9]. Alternatively, molecular fingerprints (PubChem, ECFP4, Avalon) offer fixed-length vector encodings that indicate the presence of specific substructures, and are widely used in both classical ML and DL applications [9]. Unlike traditional computer-aided drug design (CADD), which depends on manually engineered descriptors, molecular graphs represent molecules as atom-bond networks, enabling graph neural networks (GNN) to learn directly from both local chemical environments and global molecular topology (Figure 4). This approach preserves molecular symmetry and enhances performance in property prediction and 3D conformer generation, particularly when using spatially aware architectures such as SchNet, DimeNet, and GeoMol, as well as SE (3)-equivariant networks like GCPNet and TFN [10,11,12]. Each representation has limitations: SMILES and fingerprints lack 3D detail, graphs ignore explicit coordinates, and 3D methods demand conformer generation and higher computational costs (Table 1). Hybrid approaches that combine multiple representations can exploit complementary strengths.

Beyond molecular structures, the integration of omics data, such as gene expression and metabolomics, is increasingly crucial. High-dimensional gene expression profiles are often reduced via network- or pathway-based techniques, and subsequently analyzed using classical ML and DL models. However, achieving interpretability remains a persistent challenge. Metabolomic data, derived from nuclear magnetic resonance (NMR) or mass spectrometry (MS), capture phenotype-relevant features but are sensitive to pre-analytical variability. ML and DL approaches including partial least squares discriminant analysis (PLS-DA), autoencoders, and GNNs have shown promise in biomarker discovery and pathway inference [24,25]. However, the integration, normalization, and management of these heterogeneous data sources remain significant challenges that must be overcome to fully harness the potential of big data in this field.

## 2. Big Data Sources in Small-Molecule Drug Discovery

### 2.1. Public Chemical Databases

Small molecules constitute a significant portion (>90%) of marketed drugs. Several public repositories have become essential resources for small-molecule drug discovery. These repositories, such as PubChem, ChEMBL, and ZINC, contain millions of small molecules and bioassays, offering curated compound libraries for virtual screening (VS) and quantitative structure–activity relationship (QSAR) modeling (Table 2). These data sources provide multidimensional data (structures, assay readouts, target associations) that are growing exponentially. Other important databases for compound synthesis and screening include ChemSpider and BindingDB.

### 2.2. Public Biological and Omics Repositories

Advanced technologies are available for understanding disease and potential drug targets. Data generated from technologies such as microarrays and RNA sequencing (RNA-seq) and stored in repositories such as Gene Expression Omnibus (GEO), TCGA (The Cancer Genome Atlas), and ArrayExpress provide gene expression profiles for disease classification, target discovery, and dysregulated pathway identification. Genome-wide association studies (GWASs) link genetic variants to complex diseases, enabling systematic target nomination. Clinical Proteomic Tumor Analysis Consortium (CPTAC) proteomic and proteogenomic datasets further prioritize targets by protein abundance, modification state, and network context. Sequencing data from NGS are available in repositories such as Sequence Read Archive (SRA) and the National Cancer Institute Genomic Data Commons (NCIGDC), often used to identify risk genes. The Library of Integrated Network-Based Cellular Signatures (LINCS) L1000 repository contains information on changes in gene expression signatures of human cell lines treated with different chemical compounds. The Protein Data Bank (PDB) stores three-dimensional structures of proteins, DNA, and RNA, which are widely used to assess protein–ligand interactions. Platforms such as the Open Targets provide an integrated view of genetic, transcriptomic, proteomic, and literature evidence to nominate and prioritize targets systematically. Reactome and the Kyoto Encyclopedia of Genes and Genomes (KEGG) offer intuitive tools for the visualization and analysis of biological pathways and reactions. Specialized databases such as Tropical Disease Research (TDR) and Manually Annotated Targets and Drugs Online Resource (MATADOR) are also available.

### 2.3. Scientific Literature and Electronic Health Records (EHRs)

Structured, unstructured, and semi-structured data also constitute significant sources of information. Literature mining platforms such as PubMed and DisGeNET complement experimental data by extracting gene–disease–drug associations at scale. PubMed serves as a major repository of biomedical literature used for target identification. The arXiv preprint archive also contains over 2.4 million relevant articles as of 2024. Electronic health records (EHRs) represent another modality contributing to data variety.

### 2.4. Proprietary and In-House Data

Pharmaceutical companies generate and maintain large internal datasets. Companies such as Pfizer, Bayer, Novartis, Merck, and Boehringer Ingelheim maintain proprietary datasets, which are generally larger than public datasets for endpoints such as absorption, distribution, metabolism, excretion, and toxicity (ADMET) properties. These datasets include data from Investigational New Drug (IND) applications, laboratory notebooks, and internal documentation accumulated over time.

### 2.5. High-Throughput Screening (HTS) and Novel Data

High-throughput screening (HTS) platforms generate millions of measurements per experiment, contributing significantly to data volume [26,27]. DNA-encoded libraries (DEL) can generate billions of data points related to the binding of small molecules to protein targets, providing massive datasets as input for machine learning models. Automated synthesis and screening pipelines contribute to the velocity of data generation. Real-time patient monitoring, wearable devices, and ongoing clinical trial readouts also contribute to the rapid accumulation of data.

### 2.6. Generated Data from Virtual Screening (VS)

Computational approaches such as virtual screening explore vast chemical spaces by assessing extremely large libraries of compounds using computer programs, which can evaluate billions of molecules. This process generates large datasets of predicted interactions and properties.

## 3. Classical ML Models

A rich repertoire of classical ML algorithms continues to play a vital role in small-molecule drug discovery, particularly in target identification, hit identification, quantitative structure–activity/property relationship (QSAR/QSPR) modeling, and ADMET prediction (Table 3). These methods include supervised regression models, ensemble models, kernel machines, instance-based models, probabilistic classifiers, and unsupervised algorithms that often require fewer data and offer greater interpretability than deep learning methods. Supervised regression techniques, ranging from regularized linear models (e.g., Ridge, Lasso, Elastic Net) to kernel-based support vector regression (SVR) and Bayesian Gaussian Process Regression (GPR), underpin the quantitative prediction of continuous drug-like properties (IC_50_, log P) by balancing bias–variance tradeoffs and, in the case of GPR, providing uncertainty estimates critical for active learning campaigns [28,29]. Ensemble methods, particularly random forest (RF) and gradient-boosting algorithms such as XGBoost, have become de facto standards for both classification and regression QSAR tasks due to their resilience against noisy, high-dimensional chemical descriptors and their built-in feature importance measures, as exemplified by highly accurate anti-malarial efficacy models (AUC > 0.99) and kinase inhibitor discovery workflows [30]. Kernel Ridge Regression (KRR) and SVM extend this flexibility to small or imbalanced datasets by exploiting the kernel trick for nonlinear boundary and response modeling, a strategy that has successfully identified potent VEGFR2 inhibitors from HTS libraries and predicted Polo-like kinase1 (PLK1) inhibitor activities under low data conditions [31]. Probabilistic approaches such as naive Bayes (NB) and Bayesian networks (BNs) offer ultrafast training and inference for early-stage virtual screening (VS) and toxicity prediction, though their simplifying independence assumptions can limit predictive accuracy unless carefully tuned. Instance-based models such as k-nearest neighbors provide the intuitive similarity-driven ranking of compounds and have enriched activities in top-percentile screening hits, albeit at the cost of increased prediction latency and sensitivity to high-dimensional spaces. Unsupervised methods (PCA), clustering, and self-organizing maps (SOMs) remain indispensable for visualizing and organizing vast chemical libraries, guiding scaffold-hopping and structure–activity relationship (SAR) landscape exploration prior to supervised modeling [32]. Finally, multitask and transfer learning paradigms share information across related bioassays to boost performance on scarce-data targets and enable one-shot QSAR on novel scaffolds, demonstrating sub-nanomolar predictive accuracies with minimal proprietary training data [33]. Collectively, this spectrum of classical ML approaches furnishes drug discovery pipelines with robust, interpretable, and computationally efficient tools that continue to serve as benchmarks and complements to emerging deep learning frameworks (Table 3).

## 4. Deep Learning Models

Deep learning (DL), as a subfield of artificial intelligence (AI) and machine learning (ML), employs multilayered artificial neural networks (ANNs) to extract complex patterns and relationships from data, revolutionizing various stages of drug discovery and development. These methods are particularly well suited for handling the high-dimensional and diverse big data datasets generated in modern drug research (Appendix A). The key deep learning methodologies and architectures used across the drug discovery pipeline are described in the following subsections.

### 4.1. Artificial Neural Networks (ANNs)

These are foundational deep learning structures consisting of multiple layers of interconnected nodes. ANNs have experienced a resurgence due to their ability to automatically extract features from input data and capture nonlinear relationships (Figure 5). They are considered digital model brains due to their capacity for complex analysis and nonlinear relationships. They are highly recognized among deep networks and their applications in molecular modeling and pharmaceutical sciences have established a trend by providing high reliability. They have been used for pattern identification and can serve as engines. Specific uses include high-throughput screening (HTS) assays, ADMET, QSAR, pharmacophore analysis, pose validation, and lead compound formulation and development. DeepChem is an open-source tool that utilizes multitask deep ANNs for ligand screening, showing that multitask ANN can outperform standard ML methods such as random forest (RF) by synthesizing information from distinct sources. DeepTox is another example of a multitask ANNs used for toxicity prediction [47].

### 4.2. Deep Neural Networks (DNNs)

These networks have multiple hidden layers and are particularly advantageous due to their representation learning ability, allowing them to automatically perform feature extraction from input (molecular strings or graphs), obviating the need for manual feature engineering. DNNs can predict molecular properties and are used for various tasks including drug-likeness prediction, de novo molecular design, ligand–protein interaction prediction, and reaction/retrosynthesis route predictions [17]. A multitask DNN has been developed and combined with consensus modeling for large-scale QSAR predictions, improving accuracy. DNNs are also applied for ADMET analysis and in lead optimization.

### 4.3. Convolutional Neural Networks (CNNs)

Originally designed for computer vision and image processing tasks, convolutional neural networks (CNNs) use convolution kernels to recognize patterns irrespective of their location in the input, a property known as spatial equivariance. This makes them applicable to drug discovery, where molecules can be described as graphs. CNNs have been used to predict protein–ligand binding affinities, showing higher accuracy than conventional methods. They can examine trajectories generated by molecular dynamics simulations to monitor structural shifts and binding events. CNNs, including graph-based convolutional neural networks, have also been applied to predict properties such as aqueous solubility. AlphaFold, an AI-based tool for protein structure prediction, utilizes a CNN in its first step. CNNs extend traditional 2D image recognition to volumetric representations of protein–ligand complexes, applying 3D convolutional filters over voxelized grids that encode atom types and densities. Tiwari et al. demonstrated KDeep, a CNN that predicts binding affinities with markedly higher enrichment factors than empirical scoring functions, achieving up to 30% improvement in virtual screening tasks [48]. Subsequent work on CrossDocked datasets confirmed that grid-based CNNs can robustly select native-like binding poses and rank actives more accurately than classical docking pipelines, reducing false positives in hit lists [49].

### 4.4. Recurrent Neural Networks (RNNs)

Recurrent neural networks (RNNs) have emerged as powerful tools in small-molecule drug discovery, particularly for modeling and generating chemical structures encoded as sequences. Unlike feedforward neural networks, RNNs are explicitly designed to handle sequential data by maintaining a hidden state that captures information from previous time steps. This makes them particularly well suited for processing SMILES strings, which represent chemical molecules as sequences of characters. By learning the syntactic and chemical patterns within SMILES sequences, RNNs can generate novel molecular structures that retain the chemical validity and desired properties of known compounds. One of the earliest and most notable applications of RNNs in de novo drug design is DESMILES, which employs RNN architectures to generate libraries of small molecules that are chemically similar to a reference ligand. The model is trained on a large corpora of drug-like SMILES strings and, once optimized, can produce novel analogs by sampling from the learned chemical space. This approach allows the generation of focused molecular libraries with potential biological activity, enhancing lead discovery and scaffold hopping efforts.

To address the limitations of RNNs, particularly their difficulty in capturing long-range dependencies and vanishing gradient problems, more advanced architectures such as long short-term memory (LSTM) networks and gated recurrent units (GRU) have been widely adopted. Both LSTM and GRU architectures introduce gating mechanisms that regulate the flow of information through the network, enabling the model to retain and prioritize relevant sequential features over longer contexts. LSTM networks utilize a set of input, forget, and output gates to manage memory cell states, allowing the model to selectively remember or forget past information as needed. This architecture is particularly effective for learning complex syntactic and structural patterns in SMILES sequences, such as ring closures, branching, and stereochemical annotations, which require long-range dependency tracking. On the other hand, GRUs simplify this gating mechanism by combining the input and forget gates into a single update gate, while also using a reset gate. GRUs are computationally less intensive than LSTMs and often achieve comparable performance, making them attractive for large-scale molecular generation tasks.

A prominent example that integrates RNNs into drug–target interaction modeling is DeepAffinity, which employs CNN and GRU (RNN) architectures as part of its hybrid deep learning framework for compound–protein affinity prediction. By capturing temporal dependencies in both biological and chemical sequences, DeepAffinity achieves accurate predictions of binding affinities and supports tasks such as virtual screening, target prediction, and mode-of-action (MOA) elucidation [50].

### 4.5. Graph Neural Networks (GNNs)

GNNs represent molecules as graphs where atoms are nodes and bonds are edges, using message passing layers to learn structural embeddings. This approach captures both local chemical environments and global molecular topology. They have shown improvements in predicting binding affinity and ADMET properties compared to traditional fingerprint-based models. GNNs are applied in drug–target interaction and ADME-Tox prediction, with reported 15–20% performance gains over descriptor-based RF on MoleculeNet benchmarks [51]. Large-scale GNNs such as MolGPS, pretrained on extensive datasets, outperform prior baselines on numerous ADMET tasks. Related models include directed message-passing deep neural networks and models that use graph attention mechanisms. More recent studies demonstrate that attention-based GNNs, such as graph attention networks (GATs) and relational graph attention networks (RGATs), further enhance the interpretability and prediction of mechanism-of-action, achieving receiver operating characteristic area under the curve (ROC-AUC) values > 0.90 in large-scale drug–response datasets [52].

### 4.6. Generative Models for De Novo Drug Design

Traditional structure-based and ligand-based de novo methods often struggled with generating realistic, synthesizable molecules. AI-driven methods overcome these limitations by utilizing advanced representations (SMILES, molecular graphs) and generative models such as variational autoencoders (VAEs), generative adversarial networks (GANs), adversarial autoencoders (AAEs), RNNs, GNNs and RL. Tools such as MolAICal, ReLeaSE, MolPhenix, MolGPS, and Enki exemplify this AI-augmented paradigm, enabling the efficient exploration of chemical space and the generation of novel, viable small-molecule drug candidates, although challenges in synthesis prediction and ADMET accuracy remain. Generative models, such as VAEs, GANs, flow-based models, and diffusion models, are trained on existing chemical spaces. Their primary function is to learn the underlying data distribution, enabling them to sample and generate novel molecules as if drawing from this learned probability distribution. These models are indispensable for de novo drug design, particularly when integrated with fine-tuning or reinforcement learning algorithms to guide the generation process towards specific desired properties.

**Generative adversarial networks (GANs)** involve two competing neural networks (a generator and a discriminator) and are helpful for structure-based drug discovery by generating novel molecular structures (Figure 5). Deep adversarial autoencoders are also mentioned in the context of generating new molecules in oncology. GANs consist of a generator that proposes novel molecules and a discriminator that distinguishes generated from real compounds. Guimarães et al. developed ORGAN, a GAN variant with reinforcement learning rewards for drug-like metrics, achieving a twofold increase in the proportion of valid, synthesizable structures compared to vanilla VAEs [53]. Subsequent enhancements (MolGAN) incorporate graph convolutions to directly output molecular graphs, boosting novelty and diversity in scaffold generation by 25% on drug discovery benchmarks.

**Variational autoencoders (VAEs)** are a type of generative model specifically for de novo drug design to learn continuous latent representations of molecules by encoding SMILES into latent vectors and decoding them back to valid structures. Gómez-Bombarelli et al. pioneered VAEs for de novo design, demonstrating that smooth interpolations in latent space correspond to gradual changes in physicochemical properties (log P, quantitative estimate of drug likeness [QED]), enabling the gradient-based optimization of potency and drug-likeness [54]. Recently, researchers have expanded VAE frameworks with graph-based encoders, improving reconstruction fidelity and enabling scaffold hopping across diverse chemical series [55,56]. Other mentioned deep learning paradigms include autoencoders and restricted Boltzmann machines (RBNs), with VAEs being a type of autoencoder.

**Reinforcement learning (RL)** frameworks typically involve an agent (a generative model) that interacts with an environment (a predictive model acting as a critic) to make sequential decisions. The agent learns to optimize its actions by maximizing a numerical reward signal, thereby guiding the generation or modification of molecules toward desired properties. A prominent example is the ReLeaSE method, which integrates generative and predictive deep neural networks. In this system, the generative model acts as the agent, producing novel molecules, while the predictive model serves as a critic, assigning rewards based on predicted properties, thus biasing the generation toward specific physical or biological characteristics. Olivecrona et al. applied deep Q-learning on SMILES-based RNN, achieving a three-fold increase in hit rates [57]. Hybrid actor–critic frameworks integrate on-the-fly retraining of property predictors, enabling closed-loop optimization that reduces cycle times by 50% in lead refinement campaigns [58]. Reinforcement learning platforms such as GENTRL have produced discoidin domain receptor 1(DDR1) kinase inhibitors within six months of pipeline initiation, significantly compressing traditional lead identification timelines [50]. RL fine-tunes generative models toward user-defined objectives (potency, selectivity, ADME) by rewarding desired properties. It captures long-range dependencies in molecular syntax, producing chemically valid and diverse scaffolds.

### 4.7. Transformer-Based Encoders

Transformers employ self-attention mechanisms to learn contextualized embeddings from SMILES strings or molecular graphs. ChemBERTa is a transformer pretrained on >100 million SMILES strings, which, when fine-tuned on modestly sized datasets (1000–10,000 compounds), outperformed classical ML by 10–15% across multiple property prediction tasks [9]. Additional work on MolFormer and message-passing neural network (MPNN)–Transformer hybrids shows that attention layers capture substructure importance, yielding interpretable feature attributions and enabling zero-shot generalization to novel chemotypes [59]. Differentiable docking permits the gradient-based optimization of ligand poses by making scoring functions end-to-end-trainable. Wang et al. introduced DeepRMSD and Vina, a hybrid scoring function combining root-mean-square deviation (RMSD) loss with AutoDock Vina scores in a fully differentiable framework, achieving 95.4% success on Comparative Assessment of Scoring Functions 2016 (CASF-2016) docking power benchmarks, 15% higher than Vina alone [60]. This approach enables backpropagation through docking steps, refining ligand conformations directly for energy minimization in virtual screens. Chemical language models, which generate novel molecules as text strings using deep learning, are noted as particularly successful in de novo drug design. Multitask learning, often implemented using deep neural networks, is a strategy that pools data from different sources to improve predictions across various related tasks. These deep learning models are integrated into various computational tools and platforms designed for drug discovery tasks, such as DeepChem, DeepTox, DeepDR, DeepDTA, DeepDTI, DeepAffinity, MolAICal, Prediction and Analysis of Drug Molecules and Enzymes (PADME), ReLeaSE, and the Open Drug Discovery Toolkit (ODDT) [47,61,62].

## 5. AI-Driven Applications Across the Drug Discovery Pipeline

### 5.1. Target Identification

The identification of specific biological targets (DNA, RNA, proteins, enzymes, receptors, transcription factors (TFs), ion channels, metabolites) that play a pivotal role in disease progression and can be modulated by therapeutic agents (small molecules) to elicit a desired clinical outcome is recognized as the foundational step in developing successful therapies, as an inappropriate target can lead to catastrophic losses in terms of time and resources throughout the entire pipeline. Traditional target identification workflows employ a combination of biochemical and genetic techniques to elucidate the interactions between small molecules and their targets. Key methodologies include affinity-based pull-down assays, which isolate ligand–protein complexes from cellular lysates; quantitative proteomics approaches like stable isotope labeling by amino acids in cell culture (SILAC) to map interaction partners; and genome-wide loss/gain-of-function screens, such as CRISPR-Cas9 knockout or overexpression libraries, to link gene perturbations with phenotypic outcomes [63]. These strategies have been instrumental in advancing our understanding of cellular pathways and identifying potential therapeutic targets. In silico approaches accelerate this phase by computationally employing methods that can identify targets based on criteria, such as finding ligand similarity, bioactivity, and protein–protein interactions (PPIs), reducing reliance on resource-intensive wet-lab assays (Figure 6). Chemogenomic models can predict target interactions based on the chemical structures and protein sequences of targets. Network analysis can detect PPI networks to identify key players in disease pathways and potential drug targets. By combining in silico predictions with experimental validation, researchers generate novel target hypotheses that might otherwise be overlooked.

AI and ML methods are significantly accelerating this crucial process, demonstrating the capability to reduce the time required for target identification by analyzing vast and complex biological datasets, including gene expression profiles, PPI networks, and multiomics data (genomics, transcriptomics, proteomics, metabolomics). This enables them to uncover subtle patterns and insights that are often missed by conventional data analysis, leading to the identification and validation of novel, therapeutically relevant drug targets [64,65]. AI can further assist by ranking potential targets based on a comprehensive set of metrics, including their predicted druggability, clinical relevance, specificity, safety profile, novelty, and economic potential (Figure 6). AI models such as AlphaFold2 have revolutionized the field by predicting protein three-dimensional structures with atomic-level accuracy (median backbone root-mean-square deviation [RMSD] < 1 Å), enabling high-throughput modeling of drug-target interfaces and guiding small-molecule docking campaigns [66,67,68]. Extensions like AlphaFold-Multimer predict protein complexes, while AlphaFold3 incorporates diffusion-based architectures to forecast protein–small molecule interactions, broadening the druggable proteome by >50% [66]. The emergence of domain-specific large language models (LLMs), like BioGPT, is enhancing target selection by rapidly mining and synthesizing information from extensive biomedical texts and literature [69]. Transfer learning modules pretrained on large public repositories can be fine-tuned to specific drug discovery endpoints, and integrated pipelines merge multiomic, structural, and text-derived evidence into unified predictive frameworks that substantially enhance target nomination accuracy and throughput [70].

### 5.2. Hit Discovery and Virtual Screening

Identifying small molecules that interact with a target molecule, whether sourced from natural resources (such as plants or animals), synthetic libraries, or discovered through virtual screening, is known as hit identification. HTS assays employ robotics and microplate formats (96 to 1536 wells) to screen large libraries of compounds (>100,000) against biological targets, providing direct activity readouts but requiring extensive automation, reagents, and downstream data pipelines. High-throughput nuclear magnetic resonance (NMR) screening is a useful tool for analyzing protein–ligand interactions, aiding the identification of compounds that bind to specific targets [71]. Fragment-based lead discovery (FBLD) interrogates smaller libraries (<1 kDa) via NMR, surface plasmon resonance (SPR) or X-ray crystallography to detect weak binders, which are then chemically elaborated into higher-affinity leads [72]. Affinity selection techniques such as DNA-encoded libraries (DELs) and MS-based pull-down enable the rapid identification of binders from ultra-large pools, while rule-based de novo design proposes novel scaffolds guided by structural heuristics. First coined in the late 1990s, virtual screening (VS) is a suite of computational methods that evaluates vast chemical libraries (millions to billions of compounds) to predict which small molecules are most likely to bind a biological target through docking and thus serve as potential drug candidates [73]. By performing in silico docking and similarity searches, VS complements experimental HTS, enabling researchers to focus resources on the most promising compounds and dramatically reduce both time and cost [74].

Key design considerations in virtual screening (VS) encompass several critical components that collectively determine the success of hit identification. Prior to screening, chemical libraries are curated to enrich compounds with favorable drug-like properties. This involves pre-filtering based on criteria such as solubility, permeability, and synthetic feasibility, often guided by established rules like Lipinski’s Rule of Five. The choice of scoring functions, which estimate binding affinities by evaluating interactions like hydrogen bonding, hydrophobic contacts, and electrostatics, directly influences the accuracy of compound ranking and subsequent hit rates [75]. Implementing early filtering steps to eliminate compounds with reactive groups, unfavorable physicochemical properties, or pan-assay interference compounds (PAINS) reduces false positives and focuses computational efforts on more promising scaffolds. Top-ranked hits from virtual screening undergo biochemical and cell-based assays to confirm binding affinity, activity modulation, and target engagement, thereby bridging the gap between in silico predictions and empirical efficacy. Traditional approaches include structure-based VS, which uses known protein three-dimensional structures to perform molecular docking and scoring, and ligand-based VS, which mines chemical similarity to known actives via QSAR and pharmacophore modeling [76,77].

#### 5.2.1. Structure-Based Virtual Screening (SBVS)

When high-resolution structures of target proteins are available typically via X-ray crystallography, nuclear magnetic resonance (NMR) spectroscopy, or cryo-electron microscopy (cryo-EM) SBVS becomes the method of choice [76]. The conventional SBVS workflow involves the identification of the target’s active binding sites and regions crucial for its biological activity. Next, molecular docking and virtual screening techniques are utilized to computationally test and rank potential ligand molecules based on their predicted fit and interaction with the active site. Finally, the iterative optimization of lead compounds is performed to enhance their binding affinity and interaction energy [76]. Post-screening analyses, such as consensus scoring, where candidates are selected based on the agreement of multiple scoring functions, are employed to improve enrichment and effectively identify potential drug candidates. Scoring functions rank compounds by predicted binding affinity, guiding chemists toward the most promising scaffolds [78]. Advances in ensemble docking, induced-fit algorithms, and graphics processing unit (GPU)-based free-energy perturbation methods have steadily improved SBVS accuracy, with hit rates rising from historical lows of 0.001% toward routinely achievable yields above 0.1% in large-scale campaigns [79]. Despite excellent SBVS tools to enhance drug discovery (Table 4), the approach faces critical challenges including its dependence on high-resolution experimental protein structures and, unfortunately, >80% of the human proteome remains unsolved, rigid protein assumptions that limit flexibility modeling, scalability issues in navigating the vast chemical space, bottlenecks from manual optimization, and the inherent inefficiency of traditional virtual screening workflows (Figure 7).

#### 5.2.2. Ligand-Based Virtual Screening (LBVS)

In the absence of structural information, ligand-based virtual screening (LBVS) utilizes the chemical features of known active molecules to discover analogs, operating on the principle that similar structures often yield similar bioactivities (Figure 7). This includes structural information and physicochemical properties derived from known active and inactive molecules. Key methodologies within LBVS include pharmacophore modeling, which identifies and maps the essential chemical features (hydrogen bond donors/acceptors, hydrophobic centers, ionizable groups) responsible for a compound’s biological activity; and QSAR, which develops mathematical models correlating a compound’s chemical properties (descriptors) with its observed biological activity. QSAR modeling correlates molecular descriptors such as hydrophobic surface area, electronic properties, and topological indices with measured biological activity, producing predictive equations for new compounds. Pharmacophore modeling distills the essential three-dimensional arrangement of functional groups necessary for activity into abstract templates, which are then used to screen databases for compounds matching that pattern. Similarity measurements in LBVS can be performed using various molecular descriptors, ranging from one-dimensional (1D) and two-dimensional (2D) descriptors encoding chemical nature and topological features to three-dimensional (3D) descriptors related to molecular fields, shape, and volume. LBVS approaches are particularly advantageous and widely applied in scenarios where the three-dimensional structural information of the biological target is unavailable (Table 5). By focusing on chemical rather than structural similarity, LBVS reliably identifies novel chemotypes even when the target’s structure is unknown. LBVS faces key limitations including data scarcity and quality issues, bias toward known chemical space, complex and manual feature engineering, challenges in modeling nonlinear relationships, and the limited transferability of models.

Supervised ML classifiers such as RF, SVM, and GBM trained on fingerprints predict bioactivity and ADMET endpoints with greater accuracy than linear QSAR models. Modern AI and ML methods enhance traditional molecular docking approaches by incorporating techniques such as extended connectivity fingerprints (ECFPs) and GNN, which allow for the faster and more accurate prediction of protein–ligand binding affinity. GNN operates on molecular graphs to learn atom-level and substructure interactions, scaling virtual screens to billions of compounds with high throughput. Moreover, 3D-CNN processes voxelized protein–ligand complexes to automatically extract spatial binding features, outperforming classical docking scores in pose discrimination and affinity ranking [7]. Sequence-based models, LSTM, GRU, and transformer architectures consume SMILES or protein sequences to predict bioactivity and off-target liabilities, capturing long-range dependencies missed by fingerprint methods.

#### 5.2.3. Generative Virtual Screening (GVS)

The latest frontier in generative virtual screening (GVS) harnesses generative AI to design and optimize molecules in silico, rather than exhaustively screening static libraries. Models such as MolMIM combining diffusion generative networks with physics-based priors and DiffDock (a diffusion-based docking framework accelerated on GPU) iteratively propose and score novel compounds tailored to desired ADMET and binding characteristics. Generative virtual screening (VS) reduces computational overhead by focusing on promising chemical space, accelerates cycle times (DiffDock V2 is over 6 times faster than classical docking), and yields higher-quality hits for downstream validation. Ensemble methods combining diverse ML algorithms further enhance prediction robustness and mitigate model bias. Deep docking platforms combine ML-based prefiltering with structure-based docking to screen ultra-large libraries (billions of molecules) in hours instead of months. Gentile et al.’s deep docking framework demonstrated > 100-fold speedups over conventional virtual screening while retaining >80% of top actives, facilitating the practical interrogation of giga-scale chemical spaces [80]. This approach has already yielded several lead candidates advancing into early preclinical development.

### 5.3. Lead Optimization

Lead optimization transforms initial identified hits into drug candidates with balanced potency, selectivity, efficacy, and pharmacokinetic profiles. Traditional medicinal–chemistry workflows rely on hit compounds through cycles of analog synthesis, bioassay, and SAR analysis, drawing on physicochemical intuition and simple regression or classification models. Quantitative structure–activity relationship (QSAR) tools such as ALOGPS for lipophilicity prediction and associative neural network (ASNN) models for solubility correlate molecular descriptors with assay endpoints to guide analog design [81,82]. Statistical matched molecular pair analysis (MMPA), often combined with rapid conformer generation (OpenEye Omega; v.6.0), quantifies the impact of single-atom modifications on potency and lipophilicity without exhaustive synthesis [83]. Despite their reliability on modest datasets, these approaches can require dozens of design–make–test cycles over several months. AI algorithms can evaluate the effects of modifications on various biological properties, improving aspects such as binding strength, reducing off-target effects, and optimizing absorption, distribution, metabolism, and excretion (ADME) properties (Table 6) [83]. AI also evaluates molecular stability under different physiological conditions to predict how well the drug will function in the body. This automation and predictive power significantly accelerate a phase that traditionally relies heavily on human expertise and intuition.

### 5.4. ADMET Prediction

ADMET (Absorption, Distribution, Metabolism, Excretion, and Toxicity) properties are critical components of pharmacokinetics, describing how a drug is processed in the body and influencing its efficacy and safety. Optimizing these properties is essential to reduce the likelihood of nonviable molecules that either do not fall within acceptable ranges or are too rigid to be optimized. AI and ML have emerged as promising approaches for the early screening and optimization of ADMET properties, offering significant improvements in efficiency and accuracy. Platforms such as Aurigene.AI “URL https://www.aurigeneservices.com/” (accessed on 10 May 2025) have utilized ML to develop highly accurate ADMET prediction models using trusted datasets, assisting in the prioritization of hits, hit-to-lead optimization, and lead optimization. Automated ML methods are also employed to facilitate in silico ADMET property prediction, automatically searching for optimal combinations of algorithms and hyperparameters.

### 5.5. Drug Repurposing

Drug repurposing, or repositioning, involves finding new therapeutic applications for existing drugs. AI excels in this area by analyzing vast datasets, including clinical trial results, scientific literature, and genetic information, to identify potential new targets or indications for existing medications. This approach can significantly save time and costs associated with traditional drug discovery, as existing drugs have already undergone extensive safety testing. For example, BenevolentAI rapidly identified baricitinib as a potential COVID-19 treatment in just three days through its AI platform (Table 6).

### 5.6. Clinical Trial Design and Optimization

AI holds substantial potential for improving clinical trials, a phase notorious for its high costs and failure rates. AI can enhance patient recruitment and site selection by evaluating EHR and other data sources to quickly assess patient eligibility and ensure suitable candidates are screened for trials. Predictive modeling, utilizing historical data and patient characteristics, allows AI to simulate different trial designs, including dose escalation and toxicity prediction, identifying designs with the highest likelihood of success. AI tools can optimize trial protocols by adjusting variables such as dosage and treatment duration. Innovations such as synthetic control arms and digital twins can further reduce logistical and ethical challenges by simulating outcomes using real-world or virtual patient data.

### 5.7. AI in Antibiotic Discovery and Resistance Prediction

Antibiotic resistance has emerged as one of the most critical global health threats of the 21st century. Traditional approaches for discovering novel antibiotics and evaluating resistance mechanisms are often time-consuming, costly, and increasingly ineffective due to the rapid evolution of resistant pathogens. Artificial intelligence (AI) offers transformative potential by enabling high-throughput screening, resistance prediction, and the de novo design of antimicrobial compounds. AI models are now integral to antibiotic discovery pipelines, particularly in identifying novel scaffolds and the prediction of antimicrobial activity.

A generalized AI pipeline for antibiotic discovery and resistance assessment typically comprises several key stages. The process begins with data acquisition, where diverse datasets are integrated. These include genomic databases (e.g., NCBI GenBank, PATRIC) that provide microbial genetic information; chemical databases (e.g., PubChem, ChEMBL) containing small-molecule structures and properties; and phenotypic databases (e.g., CARD, ARDB), which offer experimentally validated antibiotic resistance profiles and minimum inhibitory concentration (MIC) values [105]. Following acquisition, raw data undergo preprocessing and feature engineering, involving tasks such as normalization, standardization, and the extraction of meaningful features. These features may include molecular descriptors, protein sequence motifs, or single-nucleotide polymorphisms (SNPs) associated with resistance. The curated datasets are then used for model training and optimization, wherein various AI/ML algorithms are applied. Deep learning models such as convolutional neural networks (CNNs) for predicting compound activity, SVMs for resistance classification, and RF for target identification are commonly employed (Figure 8). Trained models are used for virtual screening, enabling the rapid identification of potential antibiotic candidates from large chemical libraries based on predicted activity against bacterial targets. Simultaneously, resistance prediction models utilize microbial genomic and phenotypic data to forecast mechanisms of resistance such as efflux pump overexpression or enzymatic inactivation and to predict susceptibility profiles.

In the de novo drug design stage, generative AI models such as generative adversarial networks (GANs) and variational autoencoders (VAEs) are employed to design novel compounds with optimized physicochemical and ADMET properties. These candidate molecules, identified through virtual screening or de novo design, are subjected to experimental validation via in vitro and in vivo assays, including antimicrobial susceptibility testing (AST), cytotoxicity evaluations, and animal models. Insights obtained from experimental validation feed into model refinement and iterative optimization, forming a closed-loop system that continually enhances the predictive performance and generalizability of ML models. This iterative process significantly accelerates the discovery of effective antibiotics and improves the accuracy of resistance assessments, ultimately supporting global efforts to combat antimicrobial resistance (AMR).

Moreover, AI is increasingly used for the real-time surveillance of AMR. Platforms such as Pathogenwatch integrate genomic data with geographic and temporal metadata to monitor the spread of resistance [106]. Tools like DeepARG and MEGARes predict antimicrobial resistance genes directly from genomic and metagenomic data, while ML-based classifiers support phenotypic resistance prediction by integrating genomic, transcriptomic, and proteomic features to determine pathogen susceptibility to specific antibiotics (Table 7) [107,108]. AI also facilitates the repurposing of existing drugs by analyzing large-scale databases to identify compounds with previously unrecognized antibiotic activity.

## 6. Real-World Evidence (RWE) and Case Studies

### 6.1. Accelerated Timelines and Improved Success Rates

The integration of AI into drug discovery has led to quantifiable accelerations in development timelines and improved success probabilities. AI-discovered drugs in Phase I clinical trials have shown better success rates compared to traditionally discovered drugs, with estimates ranging from 80 to 90% for AI-driven drugs versus 40 to 65% for drugs discovered via traditional methods. This significantly improves the probability of success for new drugs, accelerating their development and ultimately delivering life-saving treatments to patients faster. AI significantly accelerates the drug discovery process by automating and optimizing various stages, reducing the time required to bring new drugs to market. The average R&D investment for a new product, which can exceed USD 2.5 billion, is also being addressed by AI’s ability to optimize resource allocation and minimize unnecessary expenditures [109].

### 6.2. Case Studies: Notable AI-Designed Drugs in Clinical Development

Pioneering efforts in applying generative AI on a large scale to drug discovery emerged around 2017. The AI-powered virtual screening of roughly 10 million compounds against the immune checkpoint protein CTLA-4 yielded several submicromolar leads that not only bound with high affinity, but also modulated T-cell activation in cellular assays, demonstrating that DL enhanced docking and ML rescoring can traverse massive chemical spaces far more rapidly than traditional methods [110]. Using generative tensorial reinforcement learning (GENTRL), Insilico Medicine designed six novel inhibitors of the discoidin domain receptor 1 (DDR1) kinase within just 21 days, four of which exhibited nanomolar potency in biochemical assays and two showed cellular activity, with lead candidates also displaying favorable mouse pharmacokinetics [111]. In the antibiotic discovery arena, MIT scientists trained a DNN on over 100 million molecules to predict antibacterial activity, leading to the identification of halicin, a compound that kills a broad spectrum of drug-resistant bacteria and uniquely prevents resistance onset in *Escherichia coli* [112]. More recently, a generative AI framework applied to BACE1 produced novel scaffolds with in silico-predicted potency against Alzheimer’s disease, representing the first AI-only de novo discovery of β-site amyloid precursor protein cleaving enzyme 1 (BACE1) inhibitors [113].

DSP-1181, an AI-designed drug developed by Sumitomo Dainippon Pharma and Exscientia, has entered clinical trials with a discovery phase taking just 12 months, a fraction of the typical 4–5 years [114]. Insilico Medicine demonstrated the ability to identify new drug targets and generate candidate molecules in just 18 months, with their idiopathic pulmonary fibrosis (IPF) drug, rentosertib, receiving its official United States Adopted Name (USAN) after both target and compound were discovered using generative AI. Scientists at BenevolentAI used an AI platform to identify baricitinib as a potential COVID-19 treatment in just 2 days [115]. Recursion Pharmaceuticals, Inc. used an unbiased, ML-powered genomics screen to rapidly identify and advance REC-1245, a potential first-in-class RNA-binding motif protein 39 (RBM39) degrader targeting solid tumors and lymphoma identification to regulatory approval in under 18 months, more than twice as fast as the industry average. These diverse case studies from CTLA-4 and DDR1 to halicin, DSP-1181, INS018-055, and BACE1 inhibitors collectively illustrate how integrated AI platforms can compress discovery timelines from years to months or weeks while delivering experimentally validated leads across therapeutic areas. While these early examples highlight remarkable speed, it is important to note that some initial AI-designed drugs, such as Exscientia’s EXS-21546 and Benevolent AI’s dermatitis drug, have faced clinical setbacks or discontinuation, underscoring the ongoing learning and refinement required in this nascent field [116].

### 6.3. Collaborative Ecosystem

The rapid advancements in AI-driven drug discovery necessitate increasingly powerful computing capabilities and diverse expertise. This has fostered a collaborative ecosystem where pharmaceutical giants, biotechnology companies, and leading technology companies are partnering to utilize AI’s potential. For instance, Roche is collaborating with companies such as Amazon Web Services and NVIDIA to enhance proprietary ML algorithms and models using accelerated computing and software, thereby speeding up drug development and improving research success rates [117]. Pfizer has also been an early adopter of AI, integrating it into pharmacovigilance since 2014 and leveraging it for rapid drug development, as seen with the COVID-19 vaccine and oral antiviral treatment [118]. Each of these platforms exemplifies how AI and deep learning can transform every phase of drug development, from target discovery to trial optimization, into data-driven, scalable pipelines with higher success rates and compressed timelines (Table 8). These collaborations are crucial for combining domain-specific knowledge with advanced computational power, accelerating the pace of innovation.

## 7. Challenges and Future Perspectives

### 7.1. Data Quality and Availability

Despite its immense promise, the adoption of AI in drug development is not without unique challenges. AI models require high-quality, diverse, and well-structured datasets for training and validation. However, much of the analytical data in pharmaceutical R&D remains fragmented, siloed, inconsistent, or locked in proprietary vendor formats, hindering seamless integration and compromising model accuracy and external validity. Without clean, standardized, and well-integrated datasets, even the most sophisticated AI algorithms will struggle to deliver meaningful results (Figure 9).

### 7.2. Model Interpretability and Explainable AI (XAI)

The black box nature of many AI algorithms, particularly deep learning models, makes it difficult for scientists to interpret predictions, raising concerns about reliability and accountability for critical decisions. While AI models can achieve high levels of accuracy, their lack of transparency limits their interpretability, which is critical for regulatory approval and clinical adoption (Figure 9). Stakeholders must be able to trust and understand the outputs of AI models, especially when they inform critical decisions about patient safety and efficacy. This necessitates innovative solutions, such as dual-model approaches and transparency tools, to balance the trade-off between high model performance and the need for explainability.

### 7.3. Ethical Considerations

The integration of AI in drug development also poses significant ethical challenges, including algorithmic bias, data privacy concerns, and the potential to exacerbate healthcare disparities. AI models trained on biased datasets may produce skewed predictions that disproportionately affect certain populations. Concerns regarding patient privacy and the necessity for stringent data governance frameworks are paramount, especially as AI increasingly utilizes patient data for personalized medicine. Ensuring fairness, accountability, and transparency in AI systems is essential to prevent discrimination and promote responsible use.

### 7.4. Regulatory Landscape

Regulatory agencies are still adapting to the rapid advancements of AI in drug development. The evolving regulatory landscape, particularly from bodies such as the Food and Drug Administration and European Medicines Agency, necessitates the development of new standards and guidelines to ensure the safety, security, and reliability of AI systems. Key areas of focus include quality assurance, premarket assessment, post-market oversight, and appropriate documentation related to data source selection and model development. International collaboration is promoted to establish global standards and best practices for AI in healthcare. The regulatory status of AI used in clinical trials, for instance, is determined by a multi-tier analysis, dependent on the software’s composition, capabilities, and specific use in the study.

### 7.5. Future Trajectory of AI in Drug Design

The future of drug design is expected to be increasingly influenced by AI-enabled platforms, combining insights from personalized medicine and nanotechnology to realize more functional and specific treatments. AI is poised to revolutionize healthcare by untangling disease biology, predicting effective approaches, and designing better therapies faster, ultimately extending and improving the lives of millions of patients. This includes the eventual realization of personalized medicine, where treatments could be optimized overnight by an AI system for an individual’s unique metabolism. The integration of quantum computing could further enhance AI’s computational capabilities, enabling faster and more precise predictions. The lab-in-a-loop strategy, where AI models are continuously refined by experimental and clinical data, represents a key future direction for continuous self-improvement in drug development. Furthermore, AI’s role is expected to expand beyond small molecules to support the design of biologics and biosimilars, such as monoclonal antibodies.

### 7.6. Federated Learning (FL)

Federated learning (FL) plays a crucial role in drug discovery by enabling collaborative data analysis without compromising privacy or security. FL was originally proposed as a solution to challenges related to data volume in AI. It was introduced specifically to preserve user privacy while benefiting from the collective knowledge of multiple data sources. This approach allows researchers to combine data from multiple organizations or institutions, creating larger datasets that can lead to better models and faster drug development. It also addresses the challenge of data scarcity in specific disease areas. FL offers several benefits beyond privacy that help address data volume challenges in AI such as reduced data transfer, scalability, real-time learning, and many others (Table 9, Figure 10).

It is evident that drug discovery is a data-intensive and high-stakes domain, where large-scale datasets ranging from genomics and proteomics to electronic health records (EHRs), real-world evidence (RWE), clinical trial data, and proprietary compound libraries are essential for developing accurate predictive models. However, these datasets are typically distributed across pharmaceutical companies, academic institutions, hospitals, and research laboratories, each bound by strict regulatory, privacy, and intellectual property constraints. FL addresses this challenge by allowing each data custodian to train a shared ML model locally on their sensitive data. Only model updates, such as weights or gradients, are transmitted to a central aggregation server, where they are combined into a global model without the raw data ever leaving the source. This paradigm shift enables unprecedented collaboration across data silos, dramatically increasing the volume and diversity of training data available for AI models. Such data richness is vital for improving the accuracy and generalizability of models used in critical drug discovery tasks such as molecular property prediction, virtual screening, biomarker identification, adverse event forecasting, and patient stratification. FL is particularly beneficial in rare disease research, where data are inherently sparse and distributed, and pooling information without violating privacy laws is crucial for progress.

Moreover, FL inherently supports learning from non-independent and identically distributed (non-IID) data—a common challenge in healthcare and biomedical data—by adapting to diverse data distributions across nodes. This capability leads to models that are more robust to variations in populations, disease subtypes, and data acquisition modalities, enhancing translational relevance. The integration of secure multiparty computation (SMPC) and differential privacy further strengthens FL’s appeal in drug discovery by ensuring that intermediate computations do not leak sensitive information. Regulatory compliance is another area where FL demonstrates strategic value. With global data protection laws such as the General Data Protection Regulation (GDPR), Health Insurance Portability and Accountability Act (HIPAA), and the emerging AI Act in the European Union, FL offers a technically sound and legally aligned method for cross-border data utilization. In doing so, it fosters global AI innovation in pharmaceuticals without compromising on privacy or ownership rights.

Overall, FL represents a paradigm shift in how ML can be applied to drug discovery. It unlocks the potential of vast, distributed biomedical datasets while preserving data sovereignty, improving model performance, and enabling ethically and legally compliant collaboration. As the industry moves toward more personalized and precision medicine approaches, FL is poised to become a cornerstone technology, accelerating the development of safer, more effective therapeutics.

## 8. Critical Perspective on the Application of AI in Small Molecule Design

Despite the remarkable advances in applying AI/ML to small-molecule drug discovery, significant limitations remain. Many studies report improvements in model accuracy or predictive performance, yet few offer deeper insights into model generalizability, interpretability, or real-world applicability. The current landscape and key technical and conceptual challenges are critically addressed and evaluated.

### 8.1. Method Comparisons and Contextual Performance

Different ML and DL approaches, such as RF, SVM, GNN, and Transformer-based architectures, are often benchmarked on narrow datasets with limited discussion on their relative applicability. While GNNs excel in learning topological and relational features from molecular graphs, they often require large, high-quality labeled datasets and suffer from over-smoothing in deeper architectures. In contrast, Transformer models demonstrate superior performance in sequence-based representations but are computationally intensive and less interpretable. The performance advantage of one model over another is often conditional on the nature of the task (e.g., QSAR vs. de novo generation), data availability, and the complexity of the molecular representation. However, such contextual dependencies are rarely addressed, resulting in misleading generalization across studies.

### 8.2. Limitations of Current Benchmarks and Datasets

While well-established benchmark resources such as ChEMBL or ZINC are invaluable, they still present several limitations: ChEMBL includes bioactivity data that vary in experimental protocols and quality, and ZINC contains synthetic accessibility artifacts. These shortcomings can inflate model performance and hinder reproducibility. Moreover, benchmark datasets often lack standardization in terms of curation, chemical diversity, and representation format (e.g., SMILES vs. InChI vs. graphs), complicating direct comparisons between models and limiting the development of universally applicable algorithms.

### 8.3. Overemphasis on Accuracy: The Blind Spot of Explainability

The dominant trend in the literature favors maximizing accuracy or AUROC scores, often at the expense of explainability and interpretability. This imbalance poses a major barrier for regulatory approval and practical integration into medicinal chemistry workflows. Few studies employ or develop methods to understand “why” a model makes given prediction, leaving behind a “black box” problem that undermines trust and usability. Without interpretable frameworks, it become difficult for domain experts to validate AI-generated molecular hypotheses or to troubleshoot when models fail.

### 8.4. Reproducibility and Generalizability Concerns

The lack of dataset standardization is a major driver of poor reproducibility across AI-driven studies. Preprocessing steps, chemical standardization, and activity thresholds vary widely, often without transparent reporting. Furthermore, many models are tested on retrospective, highly curated datasets, and their performance significantly deteriorates when exposed to prospective, noisy, or out-of-distribution data. DL models, in particular, are susceptible to overfitting and memorization, especially when trained on limited chemical spaces. This raises the question: are current models truly learning chemical principles, or are they memorizing dataset-specific patterns?

### 8.5. Disconnect from Real-World Drug Discovery

Despite hundreds of published models, only a few have made their way into actual drug discovery pipelines. A fundamental gap exists between academic benchmarks and real-world challenges, such as integrating ADMET properties, off-target effects, and synthetic feasibility into the design process. Additionally, the translational bottleneck from in silico generation to in vivo validation remains unexplored. Even for AI-generated molecules that advance to preclinical stages, many fail to meet the rigorous filters of toxicity, pharmacokinetics, or regulatory compliance. This reality underscores the need for more holistic, multi-objective optimization that balances potency, selectivity, safety, and synthesizability.

### 8.6. Toward Trustworthy and Sustainable AI in Molecular Design

For AI to become standard in small molecule design, several issues must be addressed such as bias integration through data augmentation, debiasing techniques, and cross-domain validation; interpretability frameworks that allow domain experts to interrogate model decisions; robust benchmarks with standardized data formats, curation protocols, and evaluation metrics; multi-objective learning architectures capable of optimizing complex trade-offs (e.g., efficacy vs. toxicity); and lifecycle integration, ensuring that models remain valid as new data become available. Furthermore, datasets must evolve from static repositories to dynamic, context-aware environments that incorporate metadata on assay conditions, batch effects, and experimental uncertainties. Only then can AI tools be reliably deployed in the iterative, high-stakes environment of drug discovery.

## 9. Conclusions and Future Vision

The advent of AI has mitigated several long-standing challenges in drug discovery. AI methods can manage the scale and heterogeneity of modern biomedical datasets, structural data, and literature-mined networks to uncover latent associations among genes, proteins, and pathways and analyze vast amounts of data to identify potential drug targets. Deep generative models, often combined with reinforcement learning, now enable the intelligent exploration of chemical space, automated compound optimization, and the design of novel scaffolds with patentability potential. These advances have led to the emergence of fully integrated platforms that blend cheminformatics, bioinformatics, and AI, allowing for end-to-end workflows from target identification to candidate selection. The convergence of AI with other emerging technologies such as quantum computing and closed-loop experimental platforms is expected to further accelerate drug discovery. Some AI-designed drugs have already entered clinical trials, underscoring the real-world impact of these technologies. The continuous refinement of molecular representations, combined with scalable training methods and human-in-the-loop design, will drive the development of increasingly personalized, efficient, and safe therapeutics. Despite this progress, several challenges must be addressed to fully realize AI’s potential. Key concerns include the need for high-quality, standardized data; the trade-off between model interpretability and performance; ethical issues such as data privacy and algorithmic bias; and the alignment of AI-based tools with regulatory expectations. Moreover, the generalizability and robustness of AI models remain active areas of research, particularly in low-data or noisy environments. In summary, the synergy between AI and human expertise is redefining the boundaries of pharmaceutical innovation, transforming vast, multimodal datasets into actionable insights and promising a more adaptive and effective future for drug discovery.

## Figures and Tables

**Figure 1 ijms-26-06807-f001:**
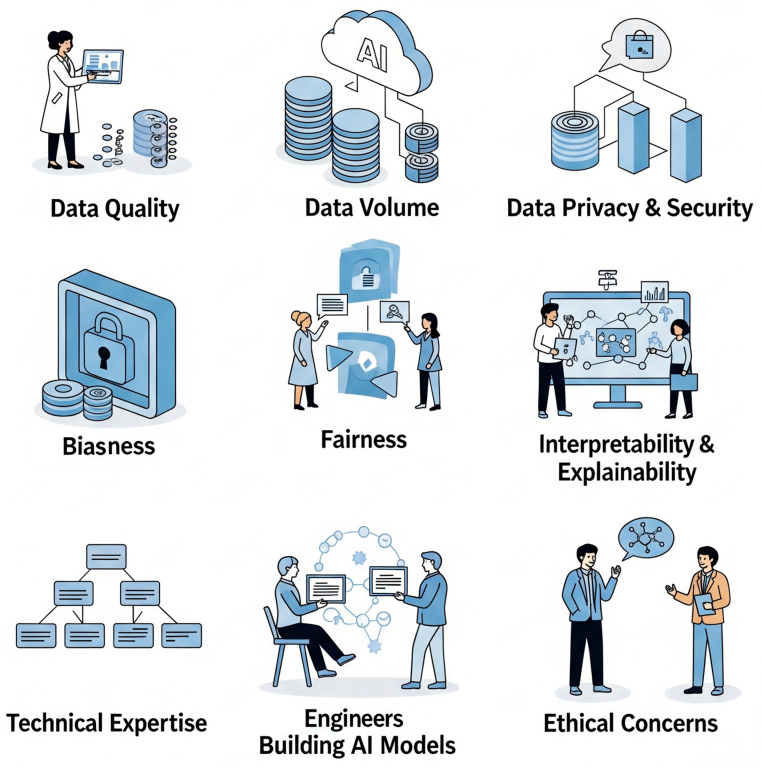
Major dimensions of artificial intelligence (AI) and its roles.

**Figure 2 ijms-26-06807-f002:**
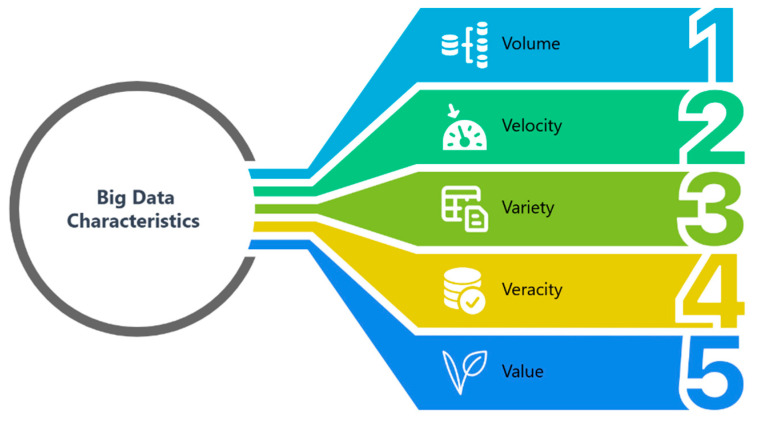
Characteristics of big data.

**Figure 3 ijms-26-06807-f003:**
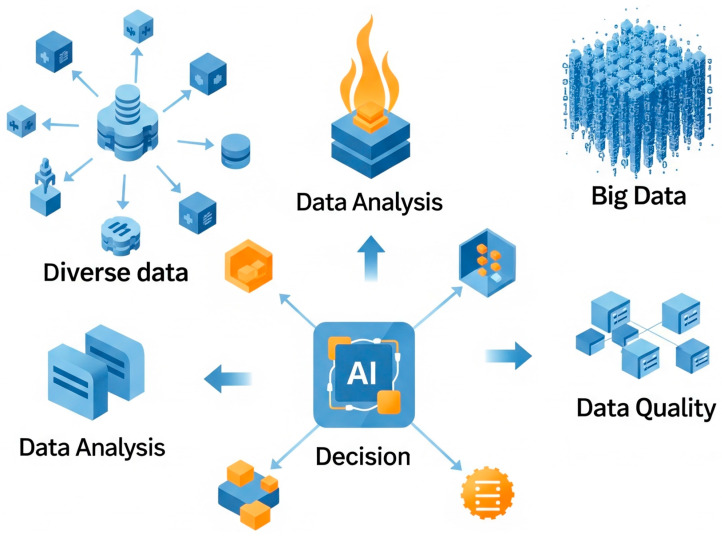
AI analysis of diverse and large-scale biomedical data enables high-quality insights to support data-informed decision making in drug discovery.

**Figure 4 ijms-26-06807-f004:**
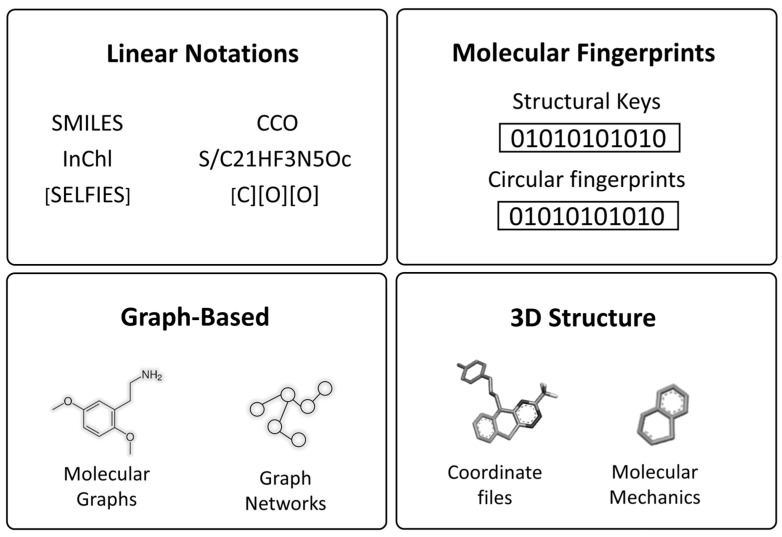
Major types of compound representations used in drug discovery.

**Figure 5 ijms-26-06807-f005:**
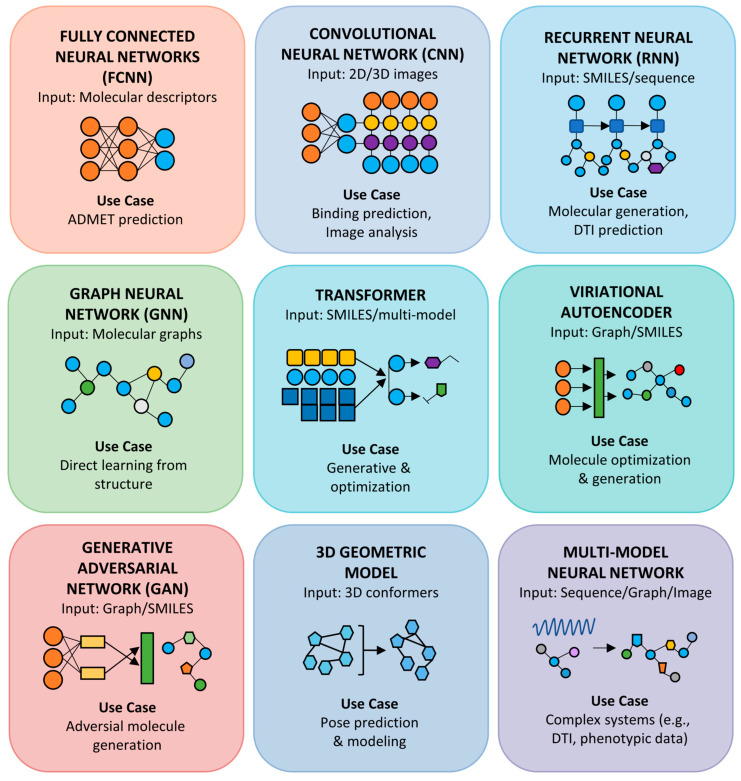
Main neural network (NN) architectures in the context of drug discovery.

**Figure 6 ijms-26-06807-f006:**
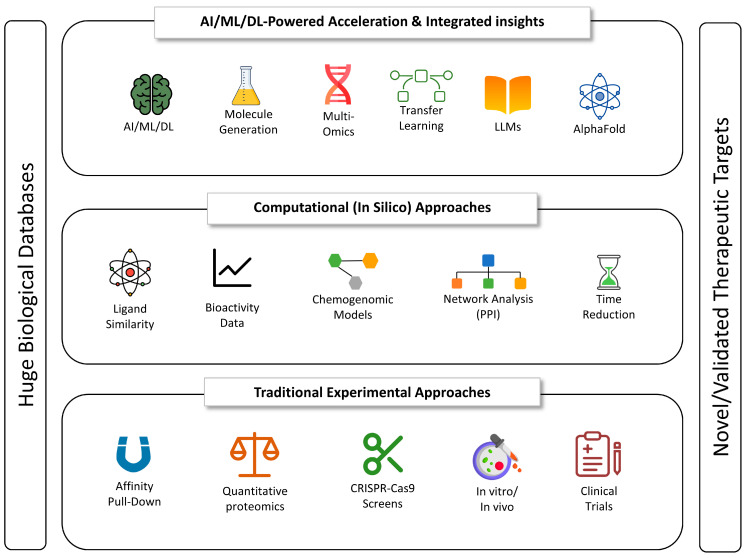
The evolving landscape of therapeutic target identification.

**Figure 7 ijms-26-06807-f007:**
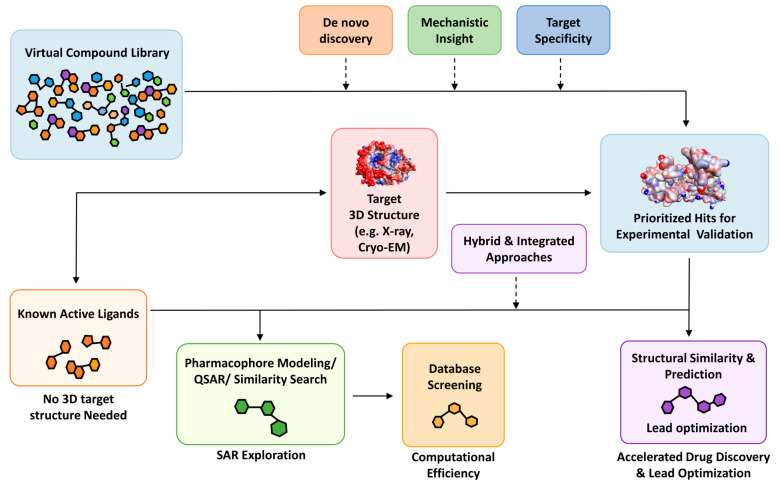
Computational strategies in small-molecule drug discovery: a comparative workflow of structure-based and ligand-based virtual screening.

**Figure 8 ijms-26-06807-f008:**
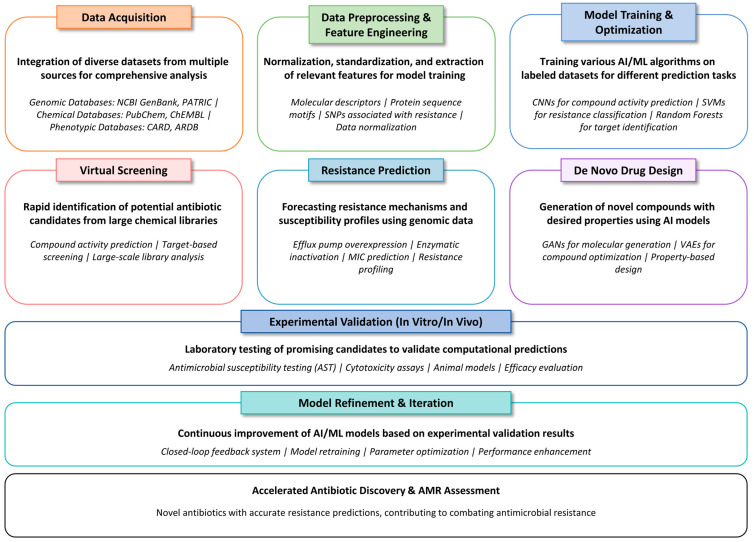
Overview of an AI-driven pipeline for antibiotic discovery and drug resistance prediction. The process begins with the integration of genomic, chemical, and phenotypic data from databases such as GenBank, ChEMBL, and CARD. After preprocessing and feature engineering, machine learning models (e.g., CNNs, SVMs, RF) are trained for virtual screening, resistance prediction, and de novo drug design using generative models (e.g., GANs, VAEs). Predicted candidates undergo experimental validation, and the results are fed back for iterative model refinement, enabling accelerated antimicrobial discovery and resistance profiling.

**Figure 9 ijms-26-06807-f009:**
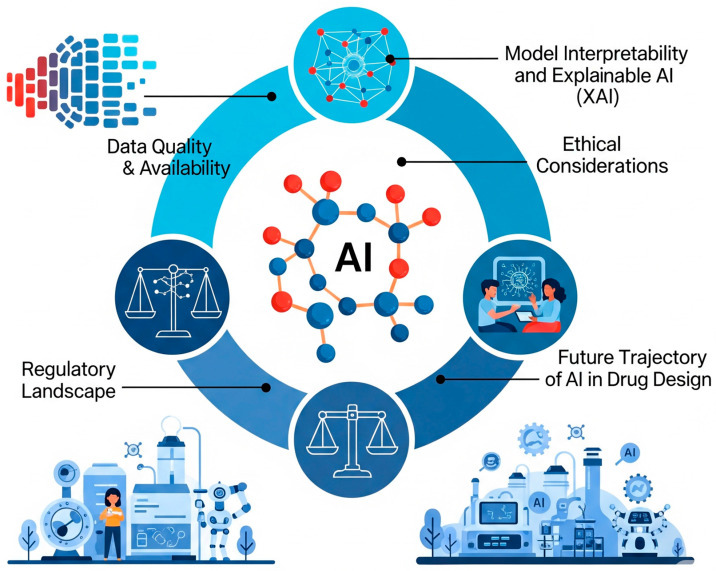
Challenges and future perspectives of AI in drug development.

**Figure 10 ijms-26-06807-f010:**
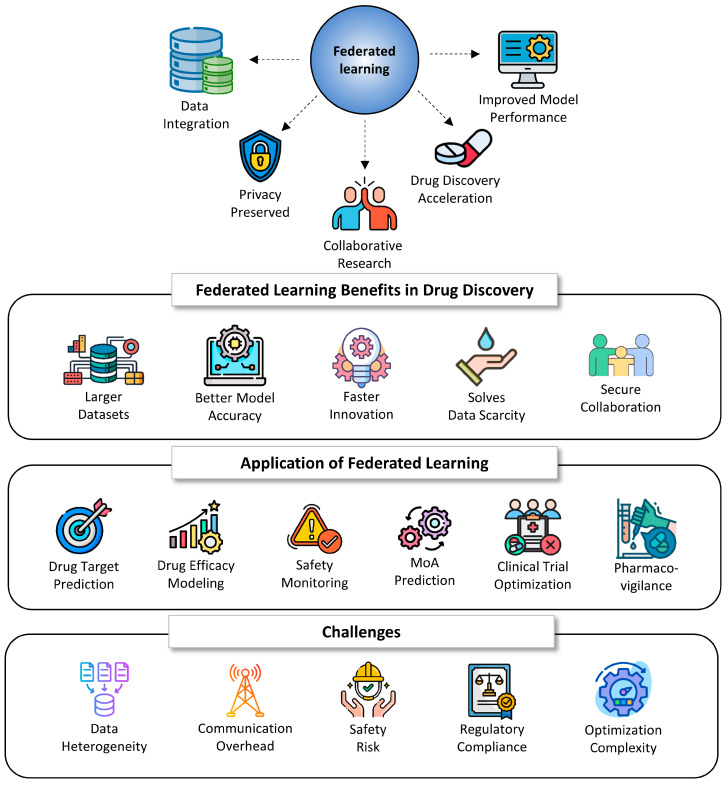
Federated learning (FL) in drug discovery: enabling secure collaborative innovation.

**Table 1 ijms-26-06807-t001:** Overview of molecular representations used in ML and DL architectures in drug discovery.

Representation Type	Classical ML	DL Architectures	Advantages	Disadvantages	Ref.
SMILES (1D strings)	SVM, RF, PLS, k-NN	RNN (LSTM, GRU), Transformers	Simple and compact, easy to store and parse, widely supported format	Non-unique representations, sensitive to syntax errors, lacks 3D stereochemical details	[13,14]
SELFIES (1D robust strings)	SVM, RF, PLS, k-NN	Transformers	100% syntactically valid, maintains expressiveness	Less human readable than smiles	[15,16]
Molecular Graphs (2D atom-bond networks)	Graph kernels, SVM, RF	MPNN, GCN, GAT	Naturally encoding of atomic connectivity, capture local and global graph topology	Computationally expensive, high memory requirements	[17,18]
3D Conformers (3D grids or point clouds)	RF, SVM	CNN, SE (3), SchNet, DimeNet, PaiNN	Encodes stereochemistry and spatial interactions	Sensitive to conformer generation, data- and computationally intensive	[19]
Fingerprints (fixed-length vectors, ECFP, MACCS)	SVM, RF, PLS, k-NN	MLP	Fast similarity search, interpretable binary features, Compact, fixed-length vectors	Ignore 3D detail, lose stereochemical and spatial details	[20]
Gene Expression Profiles	SVM, RF, LR, k-NN, PCA, PLS,	DNN, GNN, Autoencoders	Captures cellular state and pathway-level insights, reflects co-expression and regulatory activity	High dimensionality vs. small sample size, prone to overfitting, interpretability limits	[21]
Metabolite Profiles (NMR/MS)	PLS-DA, RF, SVM, PCA	Autoencoders, GNN, CNN	Biochemical and phenotype context, reflect pathway-level function	Instrument and batch variability, uneven metabolite coverage	[22,23]

**Table 2 ijms-26-06807-t002:** Public databases for ML and DL applications in structural bioinformatics and chemoinformatics for drug discovery.

Database (URL)	Scope	Key Statistics	Data Types	Classical ML	Deep Learning	Advantages	Disadvantages
ChEMBL https://www.ebi.ac.uk/chembl/ (accessed on 7 April 2025)	Curated bioactivity data: molecules, assays, mechanism of action, PD/PK	2.5 M compounds, 1.7 M assays, 16 K targets	2D structures, SMILES, InChI, physicochemical and bioactivity values (IC_50_/K_i_/EC_50_)	PLS, RF, SVM	RNN, GNN, MPNN, GCN, Transformers,	High-quality curation, integrated chemical–biological data	Drug-like chemical bias, sparse 3D coverage, assay heterogeneity
PubChem https://pubchem.ncbi.nlm.nih.gov/ (accessed on 7 April 2025)	Open chemical resource for chemical structures, properties, bioassays, and literature	119 M compounds, 330 M substances, 297 M bioactivities	Physicochemical properties, SMILES, InChI, bioassay results, substances metadata	k-NN, RF, SVM	Transformers, GNN	Largest public chemical repository, rich with biological links	Variable data quality, inconsistent annotations, extensive HTS noise
DrugBank https://go.drugbank.com/ (accessed on 7 April 2025)	Drug-centric database integrating chemistry, pharmacology, mechanisms, interactions, and ADMET	17 K drug entries, 2991 approved drugs, 1726 approved biologics	SMILES, InChI, ADMET, pharmacodynamics/kinetics, drug-target interactions	PLS, RF, SVM	GNN, transformers	Deep integration of chemical and clinical information, curated manually	Pharmaceutical bias, licensing restrictions for some applications
DrugMatrix https://cebs.niehs.nih.gov/cebs/paper/15670 (accessed on 7 April 2025)	Toxicogenomic data from rodent models, including gene expression and pathology endpoints	600 chemicals, Thousands of transcriptomic and pathology measurements	Microarray/RNA-seq gene expression, histopathology, clinical chemistry	SVM, RF, LR	DNN, autoencoders	Multimodal toxicological endpoints, standardized in vivo studies	Limited chemical diversity, rodent-specific applicability
BindingDB https://www.bindingdb.org/ (accessed on 7 April 2025)	Protein–ligand binding affinities for SAR, docking, and thermodynamics	3 M affinity data, 1.3 M compounds, 9.5 K targets	Binding constants (Kd, Ki, IC_50_), ligand structures	RF, SVM	GNN, Siamese networks	High-quality affinity data, valuable for docking and SAR benchmarking	Limited to known targets, assay variability and noise
ZINC15 https://zinc15.docking.org/ (accessed on 7 April 2025)	Ready-to-dock compound library for virtual screening	200 M molecules, 750 M purchasable compounds 37 B catalog size	2D/3D small-molecules, vendor catalogs	k-NN, RF	GNN, CNN	Ultra-large-scale screening, diverse and purchasable compounds	No bioactivity data, requires significant indexing and storage
RCSB PDB https://www.rcsb.org/ (accessed on 7 April 2025)	3D structures of macromolecules (proteins, RNA, complexes)	236 K experimental structures, 1.06 M CSM	Atomic 3D coordinates (PDB/mmCIF), electron density, ligands	RF, SVM	CNN, SE (3)-equivariant nets	High-resolution structural gold standard, interactive visualization tools	Structural bias (e.g., solubility), conformational variability
UniProt https://www.uniprot.org/ (accessed on 7 April 2025)	Protein sequences and functional annotations	252 M sequence entries, 573 K Swiss-Prot reviewed	FASTA sequences, GO terms, domains, PTMs	SVM, RF	Protein Transformers	Extensive coverage, manual curation in Swiss-Prot	Lower annotation quality in TrEMBL, redundancy
GEO https://www.ncbi.nlm.nih.gov/geo/ (accessed on 10 May 2025)	Gene expression datsets (microarray and RNA-Seq) across conditions and organisms	7.8 M samples, 4 K Datasets, 27 K platforms	Expression matrices, sample phenotypes, metadata (GSE/GSM/GPL)	SVM, RF, LR, PCA/PLS	DNN, autoencoders	Broad context diversity, MIAME-compliant standardization	High dimensionality (p ≫ n), batch effects, requires preprocessing
TCGA (GDC) https://portal.gdc.cancer.gov/ (accessed on 10 May 2025)	Pan-cancer multiomics data and clinical metadata	45 K patient cases, 1.1 M files >33 tumor types	Genomics, RNA-Seq, methylation, CNVs, proteomics, clinical annotations	SVM, RF, PLS-DA for biomarkers	DNN, GNN autoencoders	Large, deeply annotated cohorts across multiple cancers	Limited access controls, cross-platform heterogeneity
HMDB https://www.hmdb.ca/ (accessed on 10 May 2025)	Human metabolome: structures, biofluid concentrations, spectra, pathways	220 K metabolites, 5700 MS/MS spectra, 1 K NMR spectra	Structures, biofluid levels, pathway and spectral data	PLS-DA, RF, SVM	GNN, Autoencoders	High-quality curation, spectral data, physiological relevance	Human-centric bias, delayed updates, clinical complexity
ArrayExpress https://www.ebi.ac.uk/arrayexpress/ (accessed on 10 May 2025)	Functional genomics data: expression profiling, microarrays, sequencing	79 K experiments, 1.5 M profiles	Raw/processed data, experimental metadata	k-means/hierarchical, PLS-DA	DNN, GNN	Rich experiment metadata, linked to ENA	Inconsistent data formats, evolving standards, API complexity
GWAS Catalog https://www.ebi.ac.uk/gwas/ (accessed on 10 May 2025)	Curated genome-wide association studies and SNP-trait associations	7 K publications, 799 K SNP-trait links, 118 K summary stats	Summary statistics, SNP–trait p-values, study metadata	LR on summary stats	Polygenic risk-score DL models	Manually curated associations, trait-level annotations	Limited to summary-level data, study variability and design bias
LINCS L1000 https://lincsproject.org/LINCS (accessed on 10 May 2025)	Gene expression signatures from chemical and genetic perturbations	1.678 M signatures	L1000 landmark gene expression profiles (978 genes)	SVM, RF	DNN, autoencoders	Extensive perturbation atlas, standardized expression assay	Restricted to 978 genes, imputation for rest introduces noise
DisGeNET https://www.disgenet.org/ (accessed on 10 May 2025)	Gene–disease associations (GDA) from curated and text-mined sources	1983 M associations linking 29 K genes to 42 K diseases	Gene–disease relationships, ontology mappings	Network-based random walks	GNN	Combines expert curation and literature mining	Text-mining false positives, heterogeneous evidence quality
STRING https://string-db.org/ (accessed on 1 June 2025)	Protein–protein interaction (PPI) networks based on experiments, predictions, and literature	12 K organisms, 59 M proteins, 20 B interactions	Evidence-scored PPI networks	Random walk, network propagation	GNN	Comprehensive multi-evidence associations, user-friendly portal	Indirect interactions included, experimental coverage bias
STITCH http://stitch.embl.de/ (accessed on 1 June 2025)	Protein–chemical interaction networks from multiple sources	2 K organisms, 0.5 M chemicals, 9.6 M proteins, 1.6 B interactions	Protein–chemical bipartite interaction networks	Similarity-based ML on chemical/protein profiles	GNN	Combines experimental, curated, and text-mined evidence	Noisy links from text-mining, variable confidence scores
KEGG https://www.kegg.jp/ (accessed on 1 June 2025)	Integrated genomic, chemical, and pathway database	19 K compounds, 11 K glycans, 15 K reaction 8.2 K enzyme	Metabolic/reaction pathways, enzyme, drug, disease mappings	Network-based ML (e.g., random walks, PLS)	DNN, GNN	Pathway-based integrative multiomics, manually curated maps	FTP access requires subscription, slower update cycle
METLIN https://metlin.scripps.edu/ (accessed on 1 June 2025)	Experimental MS/MS spectra for metabolite identification	960 K compounds	MS/MS spectra, neutral-loss data, precursor ions	spectral matching (cosine similarity)	CNN, deep Siamese nets	Largest public MS/MS repository, regular updates	No quantitative concentration data, preprocessing (e.g., peak picking) required
Expression Atlas https://www.ebi.ac.uk/gxa/home (accessed on 1 June 2025)	Gene and protein expression across conditions and species (baseline and differential)	66 species, 4 K studies, 159 K assays	RNA-Seq, microarray, proteomics matrices	SVM, RF, LR, PCA/PLS	DNN, autoencoders, GNN	Standardized analysis pipelines, high cross-study comparability	Batch effects, occasional metadata incompleteness
Bgee https://bgee.org/ (accessed on 1 June 2025)	Healthy baseline gene expression across tissues and species	52 species, 31 K RNA-Seq libraries 56 K unique conditions	Anatomical expression calls, ontology annotations	LR, RF on calls	DNN, autoencoders	Emphasizes healthy baselines, ontology-based integration	No disease data, limited to selected model organisms
MetaboLights https://www.ebi.ac.uk/metabolights/ (accessed on 1 June 2025)	Public metabolomics repository covering diverse platforms	26 K compounds	Raw and processed NMR/MS spectra, concentrations, pathway roles	PLS-DA, RF, SVM, PCA	Autoencoders, GNN	Platform-agnostic, widely accepted for metabolomics publication	Metadata heterogeneity, identification pipelines still evolving

**Table 3 ijms-26-06807-t003:** Classical ML methodologies for small-molecule drug discovery pipelines.

Type	Methods	Advantages	Disadvantages	Applications	Examples
Supervised	SVR (RBF kernel), Ridge, Lasso, Elastic Net, GPR	SVR captures nonlinear relationships, Ridge/Lasso/Elastic Net mitigate overfitting by regularization, GPR quantifies uncertainty	SVR requires kernel, Ridge/Lasso assume linearity, GPR scale poorly with data size	Predicting potency (IC_50_, logP), permeability (PAMPA/Caco-2), active learning in lead optimization	SVR on HIV protease inhibitors [34]; phenolics [35]
Unsupervised	PCA, k-Means, Hierarchical Clustering, SOM	Visualize chemical space, Reduces noise/dimensionality	Clustering may not reflect bioactivity, SOM requires tuning	Scaffold hopping, SAR exploration, Chemical diversity analysis	SOM for SAR maps [36]; PCA for dimensionality reduction [37]
Ensemble	RF, XGBoost, LightGBM	Robust to noise, RF require minimal tuning, High accuracy (GBM), Feature importance	Computationally intensive, Risk of overfitting without tuning	QSAR/QSPR modeling, Multi-objective scoring, Bioactivity and toxicity prediction	RF for anti-malarials [38] XGBoost for Fyn kinase inhibitors [39]; RF for QSAR [40]
Kernel-Based Methods	SVM, KRR	Effective for high dimensional data, Captures nonlinear patterns	High Computational cost, Complex hyperparameter tuning	HTS classification, toxicity profiling, low-data QSAR modeling	SVM for VEGFR2 inhibitors [41]; KRR for PLK1 inhibitors [42]
Probabilistic and Bayesian	Naïve Bayes (NB), Bayesian Networks (BN)	Fast training, handles large libraries efficiently, Bayesian networks allow causal modeling	NB assumes feature independence, BN requires expert knowledge	Early-stage virtual screening, toxicity triage, Mechanism-based interpretation	BN for QSAR interpretation [43] NB reduced PI3Kγ screening cost [44]
Instance-Based models	k-NN	Intuitive and Simple, no training phase, handles multi-class problems	Prediction slows with large dataset, Suffers from curse of dimensionality	Similarity-based screening, local SAR activity estimation	k-NN for QSAR [45]
Multitask and Transfer Learning	joint RF, BN, kernel models, GPR	Boosts low data performance, Reduces experimental needs, Ideal for rare/novel targets	Risk of negative transfer, Requires related bioassays	One-shot modeling, Cross-target predictions	One-shot GPR on GPCRs [46]

**Table 4 ijms-26-06807-t004:** Structure-based virtual screening (SBVS) tools for drug discovery.

SBVS Tools	Mechanism	URL
MtiOpenScreen *	Web-based platform using AutoDock for structure-based virtual screening (SBVS)	https://bioserv.rpbs.univ-paris-diderot.fr/services/MTiOpenScreen/ (accessed on 23 May 2025)
FlexX-Scan **	High-throughput docking tool employing incremental construction algorithms to accelerate docking	https://www.biosolveit.de/products/ (accessed on 23 May 2025)
DockM8 v1.0.3 *	Consensus scoring method combining multiple docking scoring functions to improve virtual screening accuracy	https://drugbud-suite.github.io/dockm8-web/ (accessed on 23 May 2025)
BindScope * (PlayMolecule)	Deep learning-based approach using CNNs to predict binding affinities on a large scale	https://open.playmolecule.org/landing BindScope (accessed on 23 May 2025)
GeauxDock *	Monte Carlo-based docking tool using hybrid scoring functions combining physics- and knowledge-based potentials	https://www.brylinski.org/geauxdock (accessed on 23 May 2025)
EasyVS *	Web-based tool for molecule library curation and docking-based virtual screening	https://bio.tools/easyvs (accessed on 23 May 2025)
DEKOIS 2.0 *	Provides decoy sets to benchmark and challenge VS pipelines, aiding performance assessment	http://www.dekois.com (accessed on 23 May 2025)
PL-PatchSurfer2 *	Uses 3D Zernike descriptors for local surface matching between ligands and receptor pockets	https://kiharalab.org/plps2/ (accessed on 23 May 2025)
SPOT-Ligand 2 *	Template-based screening approach enhanced by a large, diverse binding homology library	https://sparks-lab.org/server/spot-ligand2/ (accessed on 23 May 2025)
Gypsum-DL *	Open-source tool for generating 3D structures of small-molecules in various tautomeric and ionization states	https://durrantlab.pitt.edu/gypsum-dl/ (accessed on 23 May 2025)
ENRI *	Tool for selecting optimal protein conformations to enhance docking outcomes	https://github.com/fibonaccirabbits/enri (accessed on 23 May 2025)

Tools marked with an asterisk (*) are freely available and open-source; tools marked with a double asterisk (**) are commercially licensed platforms.

**Table 5 ijms-26-06807-t005:** Ligand-based virtual screening (LBVS) tools for drug discovery.

LBVS Tools	Mechanism	URL
LBS-comparison *	Performance of eleven ligand binding site prediction methods were compared	https://github.com/bartongroup/LBS-comparison
VSFlow *	RDKit-based tool for substructure, fingerprint, and shape-based ligand screening	https://github.com/czodrowskilab/VSFlow
MolProphet **	Implements 2D and 3D similarity algorithms for lead identification and profiling	https://molprophet.com/
PharmScreen **	LBVS using quantum mechanics-derived hydrophobic molecular field descriptors for 3D alignment	https://pharmacelera.com/pharmscreen/
LiSiCA *	Software for 2D/3D ligand similarity using graph-based algorithms	http://insilab.org/lisica/

Tools marked with an asterisk (*) are freely available and open-source; tools marked with a double asterisk (**) are commercially licensed platforms. URLs were accessed on 11 January 2025.

**Table 6 ijms-26-06807-t006:** Applications of ML and DL for drug discovery workflows.

Application	ML Approach	DL Approach	Tools	URL	Reference
Target Prediction	RF, SVM, LR	GNN, Transformers	SwissTargetPrediction *, OpenTargets *	https://www.opentargets.org/ http://www.swisstargetprediction.ch/	[84,85]
Hit discovery	QSAR, k-NN	ANN, CNN, GAN	MolAICal *, GENTRL *, Chemprop *	https://molaical.github.io/ https://github.com/insilicomedicine/gentrl https://github.com/chemprop/chemprop	[86,87]
Lead optimization	RF, SVM, DT	RL, DNN	ChemBERTa *, GENTRL *, DeepChem *	https://deepchem.io/tutorials/transfer-learning-with-chemberta-transformers/ https://github.com/insilicomedicine/GENTRL https://deepchem.readthedocs.io/en/latest/index.html	[88]
Docking and Scoring	Docking scores, MM/PBSA	CNN, DL scoring	Gnina, DeepDocking *, Delta ML	https://github.com/gnina/gnina https://github.com/jamesgleave/DeepDockingGUI https://yzhang.hpc.nyu.edu/Delta_LinF9_XGB/	[89]
Pose Prediction	Empirical scoring	Diffusion Models, CNN	DiffDock *, RosettaVS *	https://github.com/gcorso/DiffDock https://www.rosettacommons.org/	[90]
Ligand Binding Site Prediction	QSAR, SVM, k-NN	CNN, GNN	P2Rank *, fpocket *, PrankWeb *	https://jcheminf.biomedcentral.com/articles/10.1186/s13321-018-0285-8 https://github.com/rdk/p2rank https://bioserv.rpbs.univ-paris-diderot.fr/services/fpocket/ https://prankweb.cz/	[91,92]
Bioactivity Modeling	RF, SVM, DT	DNN, CNN	ChEMBL *, ChemBERTa *	https://www.ebi.ac.uk/chembl/ https://deepchem.io/tutorials/transfer-learning-with-chemberta-transformers/	[9]
3D Pocket Detection	Geometry, Docking	DL-based Detection	CASTp *, fpocket *, PrankWeb *	https://cfold.bme.uic.edu/castpfold/https://bioserv.rpbs.univ-paris-diderot.fr/services/fpocket/ https://prankweb.cz/	[93]
SMILES-Based Generation	Genetic Algorithms	RNN, Transformers	Scaffold Decorator *, f-RAG *	https://jcheminf.biomedcentral.com/articles/10.1186/s13321-020-00441-8 https://github.com/undeadpixel/reinvent-scaffold-decorator https://github.com/NVlabs/f-RAG	[94]
De novo Design	Genetic Algorithms	RL, VAE, GAN	GENTRL *, MolAICal *, ReLeaSE *	https://github.com/insilicomedicine/GENTRL https://molaical.github.io/ https://github.com/isayev/ReLeaSE	[86]
Property-Guided Generation	QSAR, Evolutionary Strategies	RL, Generative Models	GuacaMol *, f-RAG *	https://github.com/BenevolentAI/guacamol https://github.com/NVlabs/f-RAG	[95]
Molecular Property Prediction and DTI	RF, SVM	CNN, DNN, Transformers	Chemprop *, CE-DTI *	https://github.com/chemprop/chemprop https://github.com/catly/CE-DTI	[87]
Force Field Optimization	Parameter Fitting	ML-based Force Fields	FFAST **, DPA-2 **	https://github.com/fonsecag/FFAST https://docs.deepmodeling.org/projects/deepmd/en/latest/model/dpa2.html	[96]
Dosage Optimization	PK Modeling	MIPD Tools	CURATE.AI **, Medi-Span **	https://www.curate.ai/ https://www.wolterskluwer.com/en/solutions/medi-span/medi-span/content-sets	[97,98]
Bioactive Agent Prediction	QSAR, Similarity Search	Transformers	ChEMBL *, BindingDB *	https://www.ebi.ac.uk/chembl/ https://www.bindingdb.org/	[99]
PPI Prediction	Network Analysis	GNN, Knowledge Graphs	PPI-DrugPred *, Cytoscape (v.3.10.3) *	https://github.com/ZhangHongqi215/DrugPred https://cytoscape.org/	[100,101]
Protein Folding	Molecular Dynamics (MD)	AlphaFold, RGN	AlphaFold *, RGN *	https://alphafold.ebi.ac.uk/ https://github.com/aqlaboratory/rgn	[102]
Virtual Screening	RF, Docking	DNN, RNN, FCNN	RosettaVS *, Gnina *, PyRMD *, DeepScreening *	https://www.rosettacommons.org/ https://github.com/gnina/gnina http://deepscreening.xielab.net/ https://github.com/cosconatilab/PyRMD	[102]
QSAR Modeling	LR, DT	DNN, GNN	Chemprop *	https://github.com/chemprop/chemprop	[87]
Drug Repurposing	Similarity Search	KG, Transformers	ChatGPT Repurposing (v.3.5)*, DeepDR *	https://apps.cosy.bio/drugrepochatter/ https://github.com/user15632/DeepDR	[61]
ADMET Prediction	Ensemble Models	DNN, DeepTox	DeepTox *, Chemprop *, DeepChem *	https://deep-tox.info/ https://github.com/EpistasisLab/DTox https://deepchem.readthedocs.io/en/latest/index.html https://github.com/chemprop/chemprop	[47]
MoA Prediction	Pathway, Clustering	Federated Learning	Cell Painting ML *, FederatedMoA *	https://broadinstitute.github.io/cellpainting-gallery/overview.html https://github.com/innovation-cat/Awesome-Federated-Machine-Learning	[103]
Protein Structure Prediction	Homology, Energy Minimization	Diffusion, Transformers	AlphaFold 3 *, RoseTTAFold *, RFdiffusion *	https://alphafold.ebi.ac.uk/ https://github.com/RosettaCommons/RFdiffusion https://www.rosettacommons.org/	[104]
Generative Chemistry	Rule-Based Filters	RL, GAN, Transformers	Quantiphi **	https://quantiphi.com/	[86]
Interaction Site Detection	Geometric Heuristics	3D CNN, Attention	P2Rank *, DeepRank *	https://github.com/rdk/p2rank	[91]
Protein–Ligand Complex Modeling	Docking	SE (3)-Nets, Diffusion	RFdiffusion *, DiffDock *	https://github.com/gcorso/DiffDock https://github.com/RosettaCommons/RFdiffusion	[91]
Interpretable Affinity Prediction	Autoencoders	GCN, CNN-RNN-Attention	DeepAffinity *	https://github.com/Shen-Lab/DeepAffinity	[50]
Modeling Toolkits	Scikit-learn wrappers	PyTorch/TensorFlow wrappers	ODDT *	https://github.com/oddt/oddt	[62]

Tools marked with an asterisk (*) are freely available and open-source; tools marked with a double asterisk (**) are commercially licensed platforms. URLs were accessed on 16 March 2025.

**Table 7 ijms-26-06807-t007:** Publicly available databases for antibiotic resistance gene identification and pathogen surveillance.

Name	Description	Type	Access Link
CARD *	Curated database of AMR genes and mechanisms	Resistance genes	https://card.mcmaster.ca/ https://github.com/arpcard/rgi
ResFinder *	Tool for identifying acquired AMR genes and chromosomal mutations mediating antimicrobial resistance	Resistance gene detection	http://genepi.food.dtu.dk/resfinder
MEGARes *	Hierarchical classification of AMR genes for metagenomics	Metagenomics and AMR	https://www.meglab.org/
PATRIC *	Comprehensive bacterial bioinformatics resource	Pathogen and AMR database	https://www.bv-brc.org
ARG-ANNOT *	Annotated reference gene database for AMR genes	Resistance gene curation	https://www.mediterranee-infection.com/acces-ressources/base-de-donnees/arg-annot-2/
DeepARG *	DL-based tool to predict AMR genes from DNA/protein sequences	Resistance prediction	https://github.com/gaarangoa/deeparg
Pathogenwatch *	Surveillance platform for AMR and pathogen genomics	Genomic surveillance	https://pathogen.watch/
MLAMP **	ML tool novel antimicrobial peptides with notable antibacterial potency	Antimicrobial design	https://github.com/jkwang93/AMP-Designer

Tools marked with an asterisk (*) are freely available and open-source; tools marked with a double asterisk (**) are commercially licensed platforms. URLs were accessed on 15 April 2025.

**Table 8 ijms-26-06807-t008:** AI-driven platforms covering the drug discovery continuum from target identification to clinical trial optimization.

Platform	Description	URL
PandaOmics *	Cloud-based AI platform integrating multiomics and literature mining to prioritize novel disease targets; demonstrated by identifying 28 ALS candidates validated through Drosophila models.	https://pharma.ai/pandaomics
Open Targets **	Consortium-based resource combining genetic, transcriptomic, proteomic, and NLP-derived evidence; employs XGBoost-based L2G scoring and knowledge graphs to prioritize GWAS loci for drug targeting.	https://platform.opentargets.org/
BenevolentAI **	Utilizes a proprietary ML engine and knowledge graph to normalize and analyze scientific literature, patents, and proprietary datasets for explainable target hypothesis generation.	https://www.benevolent.com/benevolent-platform/
Recursion OS **	Integrates high-content imaging, omics, and chemical data with GNN to create maps of Biology, revealing novel targets and expediting preclinical drug candidate identification.	https://www.recursion.com/technology
AtomNet ** (Atomwise)	Deep learning platform utilizing CNN trained on extensive compound libraries for structure-based virtual screening; achieved 74% hit rate across 318 targets and identified clinical candidate REC-3565.	https://www.atomwise.com/how-we-do-it/
Chemistry42 **	Generative AI suite (VAE, GAN, RNN) integrated with physics-based modeling for de novo molecular design and scaffold optimization, part of the Insilico Medicine Pharma.AI ecosystem.	https://insilico.com/chemistry42
In Clinico **	Transformer-based ensemble model that predicts Phase II–III clinical trial success (ROC AUC = 0.88) using multimodal clinical and molecular data; validated prospectively with 79% accuracy.	https://pharma.ai/inclinico
CTO 2.0 ** (ConcertAI)	SaaS platform for oncology and hematology trials leveraging real-world EHR and claims data to optimize trial eligibility, endpoint definition, and site selection.	https://www.concertai.com/clinical-trial-optimization
Deep 6 AI **	NLP-powered platform that extracts insights from unstructured EHRs, pathology reports, and clinical notes to rapidly identify trial-eligible patients, streamlining recruitment processes.	https://deep6.ai/
Saama Technologies **	AI analytics suite for patient stratification, cross-platform data integration, and compliance monitoring, supporting enhanced recruitment and data quality throughout clinical trials.	https://www.saama.com/platform/products/data-hub/
Medidata AI ** (Trials Analytics)	Real-time predictive analytics integrated into clinical infrastructure to forecast patient enrollment, site performance, and dropout risk based on industry benchmarks.	https://www.medidata.com/en/clinical-trial-products/medidata-ai/clinical-trial-analytics/
Unlearn.ai **	Employs Bayesian time-series modeling to create digital twins of trial participants, enabling in silico simulations, adaptive trial designs, and regulatory-aligned power calculations.	https://www.unlearn.ai/
Trials.ai **	NLP and ML-driven protocol optimization tool that analyzes previous trial data to refine eligibility criteria and endpoints, reducing protocol amendments and accelerating regulatory approval.	https://trials.ai/
PhaseV **	Platform offering AI-powered dashboards for trial design, risk forecasting, site selection, and scenario analysis at the portfolio level for sponsors and CROs.	https://www.phasevtrials.com/solutions
WCG AI Solutions **	Embeds generative AI into clinical trial operations to support site feasibility, dropout risk prediction, and digital recruitment within a compliant, unified workflow.	https://www.wcgclinical.com/insights/generative-ai-the-path-to-unlocking-value/
Worldwide Clinical Trials **	CRO integrating AI-powered predictive analytics and patient segmentation tools into trial operations to optimize cohort selection and accelerate development timelines.	https://www.wct.com/

Tools marked with an asterisk (*) are freely available and open-source; tools marked with a double asterisk (**) are commercially licensed platforms. URLs were accessed on 23 February 2025.

**Table 9 ijms-26-06807-t009:** Overview of federated learning benefits, application, and challenges.

Category	Aspect	Description
Benefits	Large and Diverse Datasets	FL enables the integration of data from multiple organizations, increasing dataset diversity and size, which enhances model robustness and generalizability
	Privacy-Preserving Collaboration	FL allows for collaborative model training without sharing raw data, ensuring compliance with privacy regulations (e.g., HIPAA, GDPR)
	Improved Model Accuracy	Exposure to varied data distributions improves predictive performance and reduces overfitting
	Faster Drug Development	Leveraging distributed data accelerates target discovery, compound screening, and lead optimization
	Secure Knowledge Sharing	Institutions can share model insights without compromising proprietary data, promoting pre-competitive collaboration
	Addressing Data Scarcity	Combines small, fragmented datasets from multiple sources, enhancing modeling in rare diseases or under-researched conditions
Applications	Drug Target Identification	FL integrates genomic, proteomic, and clinical data across sources to identify and validate new drug targets
	Drug Efficacy Prediction	Models trained on federated clinical datasets can predict patient-specific drug responses, aiding precision medicine
	Drug Safety Prediction	Federated models can detect adverse events early by analyzing pharmacovigilance data from EHRs and other distributed sources
	Mechanism of Action (MoA) Analysis	Helps predict molecular interactions and mechanisms of new or repurposed drugs, supporting rational drug design
	Clinical Trial Optimization	Enhances trial design and execution by aggregating insights from multiple study sites and patient populations
	Real-Time Pharmacovigilance	Enables continuous monitoring of drug safety signals using real-world data while preserving data privacy
Challenges	Data Heterogeneity	Variability in data types, formats, and distributions across institutions complicates model training and aggregation
	Communication Overhead	Sharing frequent model updates across sites can result in high bandwidth and computational demands
	Security and Adversarial Threats	FL is vulnerable to model inversion, poisoning, and gradient leakage, requiring advanced security protocols (e.g., differential privacy, secure aggregation)
	Regulatory and Ethical Constraints	FL systems must comply with national and international data privacy laws, necessitating rigorous auditing and consent processes
	Model Convergence and Optimization	Heterogeneous data and hardware require specialized optimization strategies (e.g., adaptive FedAvg) to ensure reliable model convergence

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
