# Peer review of "Digital Alchemy: The Rise of Machine and Deep Learning in Small-Molecule Drug Discovery"

_ijms, 2025, doi:10.3390/ijms26146807_

Round 1

Reviewer 1 Report

Comments and Suggestions for Authors

The paper presents an overview of the applications of artificial intelligence and machine learning in modern drug design, with a special focus on overcoming the limitations of traditional methods. The mechanisms of action, applications in virtual compound screening, and challenges related to data quality, model interpretability, and ethical and regulatory aspects are discussed. Despite the correct description, many figures and tables, I propose a few corrections:
1. More attention should be paid and databases and programs regarding antibiotic design and assessment of microorganisms' drug resistance should be added.
2. In tables 5, 6, 7, and 8, it should be added whether individual applications, platforms, etc. are available for free or for a fee.

Author Response

Comment 1: More attention should be paid and databases and programs regarding antibiotic design and assessment of microorganisms' drug resistance should be added.

Response 1:

We thank the reviewer for this valuable suggestion. We agree that including a discussion of relevant databases significantly strengthens the manuscript.

In response, we have added a new section 5.7 (highlighted) titled as “AI in Antibiotic Discovery and Resistance Prediction” on page [28-31]. This new section provides a comprehensive overview of key resources in the field, including, e.g., the Comprehensive Antibiotic Resistance Database (CARD) and the PATRIC database. We believe this addition directly addresses the reviewer's concern and provides a more thorough resource for readers.

Comment 2:  In tables 5, 6, 7, and 8, it should be added whether individual applications, platforms, etc. are available for free or for a fee.

Response 2:

We thank the reviewer for this helpful suggestion. We have now added this information to Tables 5, 6, 7, and 8 (highlighted) which specifies whether each resource is available for free, or requires a paid license.

Reviewer 2 Report

Comments and Suggestions for Authors

The manuscript systematically reviewed the recent machine and deep learning in small-molecule drug discovery. It is very well organized and well-written, which should be helpful for a wide range of readers. I would like to strongly recommend this manuscript for publication in the International Journal of Molecular Sciences.

This review manuscript covers fascinating hot topics in drug discovery. The authors also discussed the pros and cons of current and future drug discovery. Each table contains plenty of information, such as structure-based virtual screening tools for drug discovery. The websites provided by the authors are helpful for the readers.

Author Response

Response:

We are very grateful for the reviewer's time, positive feedback and strong recommendation for publication. We sincerely thank the reviewer for encouraging and supportive comments.